# Finding and Fixing Spurious Patterns with Explanations

**Gregory Plumb**                                          *gdplumb@andrew.cmu.edu*
*CMU*

**Marco Tulio Ribeiro**                                *marcotcr@cs.washington.edu*
*Microsoft Research*

**Ameet Talwalkar**                                          *talwalkar@cmu.edu*
*CMU*

**Reviewed on OpenReview:** *https://openreview.net/forum?id=whJPugmP5I*

## Abstract

Image classifiers often use spurious patterns, such as "relying on the presence of a person to detect a tennis racket," which do not generalize. In this work, we present an end-to-end pipeline for identifying and mitigating spurious patterns for such models, under the assumption that we have access to pixel-wise object-annotations. We start by identifying patterns such as "the model's prediction for tennis racket changes 63% of the time if we hide the people." Then, if a pattern is spurious, we mitigate it via a novel form of data augmentation. We demonstrate that our method identifies a diverse set of spurious patterns and that it mitigates them by producing a model that is both more accurate on a distribution where the spurious pattern is not helpful and more robust to distribution shift.

## 1 Introduction

With the growing adoption of machine learning models, there is a growing concern about *Spurious Patterns* (SPs) – when models rely on patterns that do not align with domain knowledge and do not generalize (Ross et al., 2017; Shetty et al., 2019; Rieger et al., 2020; Teney et al., 2020; Singh et al., 2020). For example, a model trained to detect tennis rackets on the COCO dataset (Lin et al., 2014) learns to rely on the presence of a person, which leads to systemic errors: it is significantly less accurate at detecting tennis rackets for images without people than with people (41.2% vs 86.6%) and only ever has false positives on images with people. Relying on this SP works well on COCO, where the majority of images with tennis rackets also have people, but would not be as effective for other distributions. Further, relying on SPs may also lead to serious concerns when they relate to protected attributes such as race or gender (Buolamwini & Gebru, 2018).

We focus on SPs where an image classifier is relying on a spurious object (e.g., using people to detect

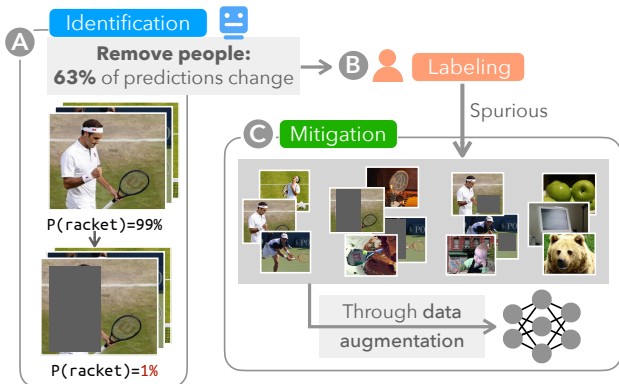

Figure 1: For the tennis racket example, SPIRE identifies this pattern by observing that, when we remove the people from images with both a tennis racket and a person, the model's prediction changes 63% of the time. Since we do not remove the tennis racket itself, we label this pattern as spurious. Then, SPIRE carefully adds/removes tennis rackets/people from different images to create an augmented training set where tennis rackets and people are independent while minimizing any new correlations between the label and artifacts in the counterfactual images (e.g., grey boxes).

tennis rackets) and we propose Spurious Pattern Identification and REpair (SPIRE)[1] as an end-to-end solution for these SPs. As illustrated in Figure 1, SPIRE *identifies* which patterns the model is using by measuring how often it makes different predictions on the original and counterfactual versions of an image. Since it reduces a pattern to a single value that has a clear interpretation, it is easy for a user to, when needed, label that pattern as spurious or valid. Then, it *mitigates* SPs by retraining the model using a novel form of data augmentation that aims to shift the training distribution towards the *balanced distribution*, a distribution where the SP is no longer helpful, while minimizing any new correlations between the label and artifacts in the counterfactual images. Each of these steps is based on the assumption that we have access to pixel-wise object-annotations to use to create counterfactual images by adding or removing objects.

To verify that the baseline model relies on a SP and quantify the impact of mitigation methods, we measure *gaps* in accuracy between images with and without the spurious object (e.g., there is a 45.4% accuracy drop between images of tennis rackets with and without people). Intuitively, the more a model relies on a SP, the larger these gaps will be and the less robust the model is to distribution shift. Consequently, an effective mitigation method will decrease these gap metrics and improve performance on the balanced distribution. Empirically, we show SPIRE's effectiveness with three sets of experiments:

- *Benchmark Experiments.* We induce SPs with varying strengths by sub-sampling COCO in order to observe how mitigation methods work in a controlled setting. Overall, we find that SPIRE is more effective than prior methods. Interestingly, we also find that most prior methods are ineffective at mitigating *negative SPs* (e.g., when the model learns that the presence of a "cat" means that there is no "tie").
- *Full Experiment.* We show that SPIRE is useful "in the wild" on the full COCO dataset. For identification, it finds a diverse set of SPs and is the first method to identify negative SPs, and, for mitigation, it is more effective than prior methods. Additionally, we show that it improves zero-shot generalization (i.e., evaluation without re-training) to two challenging datasets: UnRel (Peyre et al., 2017) and SpatialSense (Yang et al., 2019). These results are notable because most methods produce no improvements in terms of robustness to natural distribution shifts (Taori et al., 2020).
- *Generalization Experiments.* We illustrate how SPIRE generalizes beyond the setting from our prior experiments, where we considered the object-classification task and assumed that the dataset has pixel-wise object-annotations to use to create counterfactual images. Specifically, we explore three examples that consider a different task and/or do not make this assumption.

## 2 Related Work

We discuss prior work as it pertains to identifying and mitigating SPs for image classification models.

**Identification.** The most common approach is to use explainable machine learning (Simonyan et al., 2013; Ribeiro et al., 2016; Selvaraju et al., 2017; Ross et al., 2017; Singh et al., 2018; Dhurandhar et al., 2018; Goyal et al., 2019; Koh et al., 2020; Joo & Kärkkäinen, 2020; Rieger et al., 2020). For image datasets, these methods rely on local explanations, resulting in a slow process that requires the user to look at the explanation for each image, infer what that explanation is telling them, and then aggregate those inferences to assess whether or not they represent a consistent pattern (Figure 2). In addition to this procedural difficulty, there is uncertainty about the usefulness of some of these methods for model debugging (Adebayo et al., 2020; 2022).

In contrast, SPIRE avoids these challenges by measuring the aggregated effect that a specific counterfactual has on the model's predictions (e.g., the model's prediction changes 63% of the time when we remove the people from images with both a tennis racket and a person). While prior work has defined similar measurements, Shetty et al. (2019) and Singh et al. (2020) fail to identify negative SPs and Xiao et al. (2021) fail to identify specific SPs (e.g., they find that the model is relying on "something" rather than "people").

**Mitigation.** While prior work has explored data augmentation for mitigation (Hendricks et al., 2018; Shetty et al., 2019; Teney et al., 2020; Chen et al., 2020a; Agarwal et al., 2020), it has done so with augmentation strategies that are agnostic to the training distribution (e.g., Shetty et al. (2019) simply remove either the tennis rackets or the people, as applicable, uniformly at random for each image). In contrast, SPIRE aims to use counterfactual images to create a training distribution where the label is independent of the spurious

---

[1]Code is available at `https://github.com/GDPlumb/SPIRE`

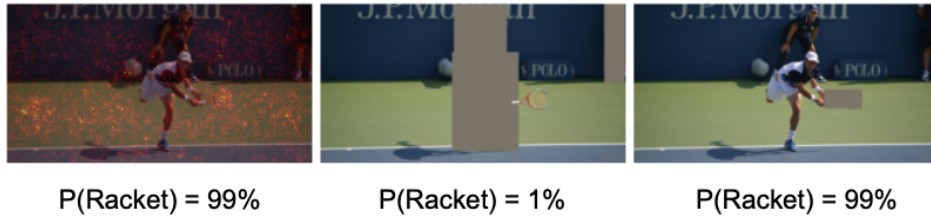

P(Racket) = 99%        P(Racket) = 1%        P(Racket) = 99%

Figure 2: Based on the saliency map (Simonyan et al., 2013) (Left), one might mistakenly infer that the model is not relying on the person. However, the model fails to detect the racket after the person is removed (Center) and incorrectly detects a racket after it is removed (Right).

object, while minimizing any new correlations between the label and artifacts in the counterfactual images. We hypothesize that this is why we find that SPIRE is more effective than these methods.

Another line of prior work adds regularization to the model training process (Ross et al., 2017; Hendricks et al., 2018; Wang et al., 2019; Rieger et al., 2020; Teney et al., 2020; Liang et al., 2020; Singh et al., 2020). Some of these methods specify which parts of the image should not be relevant to the model's prediction (Ross et al., 2017; Rieger et al., 2020). Other methods encourage the model's predictions to be consistent across counterfactual versions of the image (Hendricks et al., 2018; Teney et al., 2020; Liang et al., 2020). All of these methods could be used in conjunction with SPIRE.

Finally, there are two additional lines of work that make different assumptions than SPIRE. Making weaker assumptions, there are methods based on sub-sampling, re-weighting, or grouping the training set (Chawla et al., 2002; Sagawa* et al., 2020; Creager et al., 2020). These methods have been found to be less effective than methods that use data augmentation or regularization (Rieger et al., 2020; Neto, 2020; Singh et al., 2020; Goel et al., 2021). Making stronger assumptions, there are methods which assume access to several distinct training distributions (Wen et al., 2020; Chen et al., 2020b).

Consequently, the methods designed for image classification that use data augmentation or regularization represent SPIRE's most direct competition. As a result, we compare against "Right for the Right Reasons" (RRR) (Ross et al., 2017), "Quantifying and Controlling the Effects of Context" (QCEC) (Shetty et al., 2019), "Contextual Decomposition Explanation Penalization" (CDEP) (Rieger et al., 2020), "Gradient Supervision" (GS) (Teney et al., 2020), and the "Feature Splitting" (FS) method from (Singh et al., 2020).

## 3  Spurious Pattern Identification and REpair

In this section, we explain SPIRE's approach for addressing SPs. We use the object-classification task as a running example, where *Main* is the object being detected and *Spurious* is the other object in a SP. The same methodology applies to any binary classification problem with a binary "spurious feature;" see Appendix B.1 for a discussion on how to do this.

**Preliminaries.** We view a dataset as a probability distribution over a set of *image splits*, which we call *Both*, *Just Main*, *Just Spurious*, and *Neither*, depending on which of Main and/or Spurious they contain (e.g., Just Main is the set of images with tennis rackets, without people, and either with or without any other object). Figure 3 (Left) shows these splits for the tennis racket example. Critically, we can take an image from one split and create a counterfactual version of it in a different split by either adding or removing either Main or Spurious (e.g., removing the people from an image in Both moves it to Just Main). To do this, we assume access to pixel-wise object-annotations; note that this is a common assumption (i.e., RRR, QCEC, CDEP, and GS also make it). See Appendix B.2 for more details on the counterfactual images.

To summarize, SPIRE makes two general assumptions:

- That we can determine which of these splits an image belongs to.
- That we can create a counterfactual version of an image from one split that belongs to a different split.

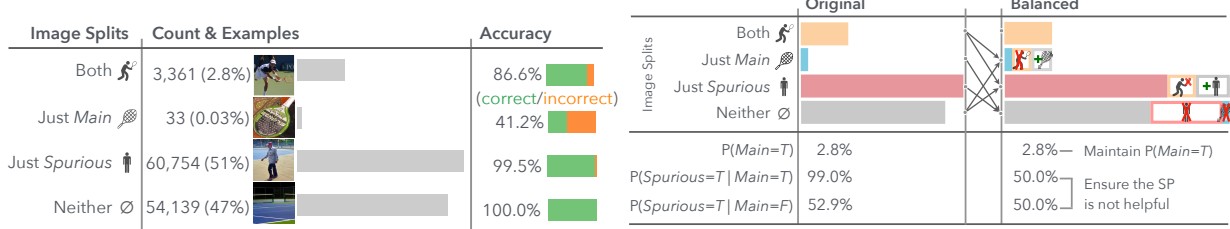

Figure 3: **Left.** The training image splits and the original training distribution for the tennis racket example. Because of the strong positive correlation between Main and Spurious, it is helpful for the model to rely on this SP. **Right.** The balanced distribution for the tennis racket example. This SP is no longer helpful because Main and Spurious are now independent and there are the same number of images in Both and Just Main.

**Identification.** SPIRE measures how much the model relies on Spurious to detect Main by measuring the probability that, when we remove Spurious from an image from Both, the model's prediction changes (e.g., the model's prediction for tennis racket changes 63% of the time when we remove the people from an image with both a tennis racket and a person). Intuitively, higher probabilities indicate stronger patterns.

To identify the full set of patterns that the model is using, SPIRE measures this probability for all (Main, Spurious) pairs, where Main and Spurious are different, and then sorts this list to find the pairs that represent the strongest patterns. Recall that not all patterns are necessarily spurious and that the user may label patterns as spurious or valid as needed before moving to the mitigation step.

**Mitigation.** It is often, but not always, the case that there is a strong correlation between Main and Spurious in the original training distribution, which incentivizes the model to rely on this SP. As a result, we want to define a distribution, which we call the *balanced distribution*, where relying on this SP is neither inherently helpful nor harmful. This is a distribution, exemplified in Figure 3 (Right), that:

- *Preserves P(Main).* This value strongly influences the model's relative accuracy on {Both, Just Main} versus {Just Spurious, Neither} but does not incentivize the SP. As a result, we preserve it in order to maximize the similarity between the original and balanced distributions.
- *Sets P(Spurious | Main) = P(Spurious | not Main) = 0.5.* This makes Main and Spurious independent, which removes the statistical benefit of relying on the SP, and assigns equal importance to images with and without Spurious. However, this does not go so far as to invert the original correlation, which would directly punish reliance on the SP.

As shown in Figure 3 (Right), SPIRE's mitigation strategy uses counterfactual images to manipulate the training distribution. The specifics are described in Section 3.1, but they implement two goals:

- *Primary: Shift the training distribution towards the balanced distribution.* While the original distribution often incentivizes the model to rely on the SP, the balanced distribution does not. However, adding too many counterfactual images may compromise the model's accuracy on natural images. As a result, we want to shift the training distribution towards, but not necessarily all the way to, the balanced distribution.
- *Secondary: Minimize the potential for new SPs.* While shifting towards the balanced distribution, we may inadvertently introduce new potential SPs between Main and artifacts in the counterfactual images. For example, augmenting the dataset with the same counterfactuals that SPIRE uses for identification (i.e., images from Both where Spurious has been covered with a grey box) introduces the potential for a new SP because P(Main | "grey box") = 1.0. Because the augmentation will be less effective if the model learns to rely on new SPs, we minimize their potential by trying to set P(Main | Artifact) = 0.5.

## 3.1 Specific Mitigation Strategies

While SPIRE's augmentation strategy follows the aforementioned goals, its specific details depend on the problem setting, which we characterize using two factors:

Table 1: Setting 1. For $p = 0.9$ and $p = 0.1$, we show the original size of each split for a dataset of size 200 as well as the size of each split after SPIRE's or QCEC's augmentation. Note that SPIRE produces the balanced distribution, while QCEC does not even make Main and Spurious independent.

| | p = 0.9 | | | p = 0.1 | | |
| Split | Original | SPIRE | QCEC | Original | SPIRE | QCEC |
|---|---|---|---|---|---|---|
| Both | 90 | 90 | 90 | 10 | 90 | 10 |
| Just Main | 10 | 90 | 55 | 90 | 90 | 95 |
| Just Spurious | 10 | 90 | 55 | 90 | 90 | 95 |
| Neither | 90 | 90 | 110 | 10 | 90 | 190 |

- *Can the counterfactuals change an image's label?* For tasks such as object-classification, counterfactuals can change an image's label by removing or adding Main. However, for tasks such as scene identification, we may not have counterfactuals that can change an image's label. For example, we cannot turn a runway into a street or a street into a runway by manipulating a few objects. This fundamentally shapes how counterfactuals can change the training distribution.
- *Is the dataset class balanced?* While working with class balanced datasets drastically simplifies the problem and analysis, it is not an assumption that usually holds in practice.

These two factors define the three problem settings that we consider, which correspond to the experiments in Sections 5.1, 5.2, and 5.3 respectively. For each setting, we summarize what makes it interesting, define SPIRE's specific augmentation strategy for it, and then discuss how that strategy meets SPIRE's goals.

**Setting 1: Counterfactuals can change an image's label and the dataset is class-balanced.** Here, $P(\text{Main}) = P(\text{Spurious}) = 0.5$ and we can define the training distribution by specifying $p = P(\text{Main} \mid \text{Spurious})$. If $p > 0.5$, SPIRE moves images from {Both, Neither} to {Just Main, Just Spurious} with probability $\frac{2p-1}{2p}$ for each of those four combinations. If $p < 0.5$, SPIRE moves images from {Just Main, Just Spurious} to {Both, Neither} with probability $\frac{p-0.5}{p-1}$.

Table 1 shows how SPIRE changes the training distributions for $p = 0.9$ and $p = 0.1$. For $p = 0.9$, it succeeds at both of its goals. For $p = 0.1$, it produces the balanced distribution, but does add the potential for new SPs because $P(\text{Main} \mid \text{Removed an object}) = 0$ and $P(\text{Main} \mid \text{Added an object}) = 1$. We contrast SPIRE to the most closely related method, QCEC (Shetty et al., 2019), which removes either Main or Spurious uniformly at random, as applicable, from each image. For both values of $p$, QCEC does not make Main and Spurious independent and adds the potential for new SPs. This example highlights the fact that, while prior work has used counterfactuals for data augmentation, SPIRE uses them in a fundamentally different way by considering the training distribution.

**Setting 2: Counterfactuals can change an image's label, but the dataset has class imbalance.** Class imbalance makes two parts of the definition of the balanced distribution problematic for augmentation:

- *Preserves P(Main).* When $P(\text{Main})$ is small, this means that we generate many more counterfactual images without Main than with it, which can introduce new potential SPs.
- *Sets P(Spurious | not Main) = 0.5.* When $P(\text{Spurious})$ is also small, this constraint requires that most of the counterfactual images we generate belong to Just Spurious, which can lead to the counterfactual data outnumbering the original data by a factor of 100 or more for this split.

Consequently, we relax these constraints. If $P(\text{Spurious} \mid \text{Main}) > P(\text{Spurious})$, SPIRE creates an equal number of images to add to Just Main/Spurious by removing the appropriate object from an image from Both. Specifically, this number is the smallest positive solution for $\delta$ to: $\frac{|\text{Both}|}{|\text{Both}|+|\text{Just Spurious}|+\delta} = \frac{|\text{Just Main}|+\delta}{|\text{Just Main}|+|\text{Neither}|+\delta}$. Otherwise, SPIRE creates an equal number of images to add to Both/Just Spurious by adding Spurious to Just Main/Neither. Specifically, this number solves: $\frac{|\text{Both}|+\delta}{|\text{Both}|+|\text{Just Spurious}|+2\delta} = \frac{|\text{Just Main}|}{|\text{Just Main}|+|\text{Neither}|}$. SPIRE achieves its primary goal by making $P(\text{Main} \mid \text{Spurious}) = P(\text{Main} \mid \text{not Spurious})$ (i.e., Main and Spurious are now independent) and it achieves its secondary goal by adding an equal number of counterfactual images with and without Main (i.e., $P(\text{Main} \mid \text{Artifact}) = 0.5$).

**Setting 3: Counterfactuals cannot change an image's label.** This constraint prevents the previous strategies from being applied. Therefore, SPIRE removes Spurious from every image with Spurious and adds Spurious to every image without Spurious. While this process does achieve SPIRE's primary goal, it does not achieve SPIRE's secondary goal (i.e., the original correlation between the label and Spurious is the same as the correlation between the label and grey boxes from removing Spurious).

## 4 Evaluation

Because relying on the SP is usually helpful on the original distribution, we cannot measure the effectiveness of a mitigation method using that distribution. Instead, we measure the model's performance on the balanced distribution, using metrics such as accuracy and average precision. Intuitively, using the balanced distribution provides a fairer comparison because the SP is neither helpful nor harmful on it. However, like any performance metric that is aggregated over a distribution, these metrics hide potentially useful details.

We address this limitation by measuring the model's accuracy on each of the image splits. These per split accuracies yield a more detailed analysis, allow us to estimate the model's performance on any distribution (e.g., the balanced distribution) by re-weighting the model's accuracy on them, and allow us to calculate two "gap metrics". The *Recall Gap* is the difference in accuracy between Both and Just Main; the *Hallucination Gap* is the difference in accuracy between Neither and Just Spurious. Intuitively, a smaller recall gap means that the model is more robust to distribution shifts that move weight between Both and Just Main. The same is true for the hallucination gap and shifts between Neither and Just Spurious. As a concrete example of these metrics, consider the tennis racket example (Figure 3 Left), where we observe that the recall gap is 45.4% (i.e., the model is much more likely to detect a tennis racket when a person is present) and a hallucination gap of 0.5% (i.e., the model is more likely to hallucinate a tennis racket when a person is present; see Appendix C.2).

Note that these per split accuracies are measured using only natural (i.e., not counterfactual) images, in order to prevent the model from "cheating" by learning to use artifacts in the counterfactual images. As a result, the gap metrics and the performance on the balanced distribution also only use natural images.

**Class Balanced vs Imbalanced Evaluation.** When there is class balance, we use the standard prediction threshold of 0.5 to measure a model's performance using accuracy on the balanced distribution (i.e., *balanced accuracy*) and its gap metrics. When there is class imbalance, Average Precision (AP), which is the area under the precision vs recall curve, is the standard performance metric. Analogous to AP, we can calculate the Average Recall Gap by finding the area under the "absolute value of the recall gap" vs recall curve; the Average Hallucination Gap is defined similarly. As a result, we measure a model's performance using AP on the balanced distribution (i.e., *balanced AP*) and the Average Recall/Hallucination Gaps. Appendix D provides more details for these metrics by showing how they are calculated using the tennis racket example.

## 5 Experiments

We divide our experiments into three groups:

- In Section 5.1, we induce SPs with varying strengths by sub-sampling COCO in order to understand how mitigation methods work in a controlled setting. We show that SPIRE is more effective at mitigating these SPs than prior methods. We also use these results to identify the best prior method, which we use for comparison for the remaining experiments.
- In Section 5.2, we find and fix SPs "in the wild" using all of COCO; this means finding multiple naturally occurring SPs and fixing them simultaneously. We show that SPIRE identifies a wider range of SPs than prior methods and that it is more effective at mitigating them. Additionally, we show that it improves zero-shot generalization to two challenging datasets (UnRel and SpatialSense).
- In Section 5.3, we demonstrate that SPIRE is effective for problems other than COCO. To do so, we consider tasks other than object-classification and/or explore techniques for constructing counterfactuals for datasets without pixel-wise object-annotations.

For the baseline models (i.e., the normally trained models that contain the SPs that we are going to identify and mitigate), we fine-tune a pre-trained version of ResNet18 (He et al., 2016) (see Appendix E). We compare

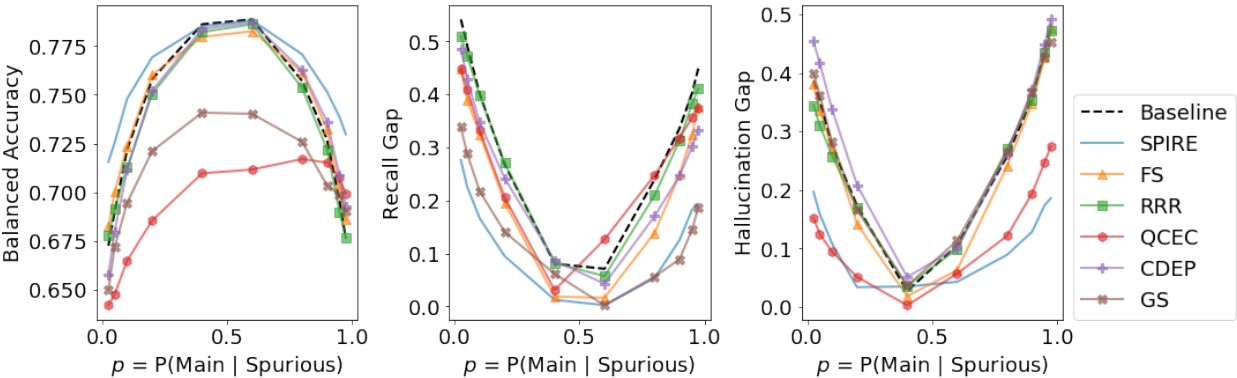

Figure 4: A comparison of the baseline model to various mitigation methods. The results shown are averaged across both the pairs accepted for our benchmark and across eight trials. **Left - Balanced Accuracy.** For $p \leq 0.2$ and $p \geq 0.8$, SPIRE produces the most accurate models. None of the methods have much of an impact for $p = 0.4$ or $p = 0.6$, likely because those create weak SPs. **Center/Right - Recall/Hallucination Gaps.** SPIRE generally shrinks the absolute value of both of the gap metrics by more than prior methods.

SPIRE to RRR (Ross et al., 2017), QCEC (Shetty et al., 2019), CDEP (Rieger et al., 2020), GS (Teney et al., 2020), and FS (Singh et al., 2020). We use the evaluation described in Section 4 and, for any split that is too small to produce a reliable accuracy estimate, we acquire additional images (using Google Images) such that each split has at least 30 images to use for evaluation.

**Measuring variance across trials.** Because we are applying these mitigation methods to the baseline model, the results of some metric (e.g., Balanced AP) for the mitigated model and the baseline model are not independent. As a result, we do not directly report the variance of this metric across trials and, instead, report the variance its difference between the mitigated and baseline models. Subsequently, the "$[\sigma=]$" entries in our tables will denote the standard deviation of a particular metric's difference across trials.

### 5.1 Benchmark Experiments

We construct a set of benchmark tasks from COCO consisting of different SPs with varying strengths, by manipulating the model's training distribution, in order to better understand how mitigation methods work in a controlled setting. Appendix F has additional details.

**Creating the benchmark.** We start by finding each pair of objects that has at least 100 images in each split of the testing set (13 pairs). For each of those pairs, we create a series of controlled training sets of size 2000 by sampling images from the full training set such that P(Main) = P(Spurious) = 0.5 and $p$ = P(Main | Spurious) ranges between 0.025 and 0.975. Each controlled training set represents a binary task, where the goal is to predict the presence of Main.

While varying $p$ allows us to control the strength of the correlation between Main and Spurious (i.e., $p$ near 0 indicates a strong negative correlation while $p$ near 1 indicates a strong positive correlation), it does not guarantee that the model actually relies on the intended SP. Indeed, when we measure the models' balanced accuracy as $p$ varies, we observe that 5 out of the 13 pairs show little to no loss in balanced accuracy as $p$ approaches 1. Consequently, subsequent evaluation considers the other 8 pairs. For these pairs, the model's reliance on the SP increases as $p$ approaches 0 or 1 as evidenced by the increasing loss of balanced accuracy.

**Results.** Figure 4 (Left) presents the balanced accuracy results. We find that SPIRE consistently improves balanced accuracy and that it does so by more than prior methods. Interestingly, while most prior methods are beneficial for strong positive SPs ($p \geq 0.9$), only FS is also (mildly) beneficial for negative SPs ($p < 0.5$).

Figure 4 (Center/Right) presents the gap metric results. We find that SPIRE is the most effective method at shrinking these metrics, which indicates that it produces a model that is more robust to distribution shift.

Table 2: A few examples of the SPs identified by SPIRE for the Full Experiment. For each pair, we report several basic dataset statistics including *bias*, $\frac{\text{P(Spurious | Main) - P(Spurious)}}{\text{P(Spurious)}}$, which captures how far Main and Spurious are away from being independent as well as the sign of their correlation.

| Main | Spurious | P(M) | P(S) | P(S \| M) | bias |
|------|----------|------|------|-----------|------|
| tie | cat | 0.03 | 0.04 | 0.01 | -0.66 |
| toothbrush | person | 0.01 | 0.54 | 0.54 | -0.01 |
| bird | sheep | 0.03 | 0.01 | 0.01 | 0.00 |
| frisbee | person | 0.02 | 0.54 | 0.83 | 0.54 |
| tie | person | 0.03 | 0.54 | 0.95 | 0.76 |
| tennis racket | person | 0.03 | 0.54 | 0.99 | 0.83 |
| dog | sheep | 0.04 | 0.01 | 0.03 | 1.05 |
| frisbee | dog | 0.02 | 0.04 | 0.24 | 5.44 |
| fork | dining table | 0.03 | 0.10 | 0.76 | 6.56 |

Table 3: Mitigation results for the Full Experiment. Balanced AP is averaged across the SPs identified by SPIRE. Similarly, the gap metrics are reported as the "mean (median)" change from the baseline model, aggregated across those SPs.

| | Original MAP | Balanced AP | %Δ Avg. Recall Gap | %Δ Avg. Hallucination Gap |
|---|---|---|---|---|
| Baseline | **64.1** | 46.2 | — | — |
| SPIRE | 63.7 [$\sigma$=0.1] | **47.3** [$\sigma$=0.5] | **-14.2 (-14.5)** | **-28.1 (-27.3)** |
| FS | 62.5 [$\sigma$=1.0] | 44.7 [$\sigma$=1.9] | 9.7 (-5.9) | 25.7 (-6.9) |

Table 4: The MAP results of a zero-shot evaluation on the classes that are in the UnRel/SpatialSense datasets that SPIRE also identified as being Main in a SP.

| | UnRel | SpatialSense |
|---|---|---|
| Baseline | 38.9 | 20.3 |
| SPIRE | **41.3** [$\sigma$=1.3] | **20.7** [$\sigma$=0.5] |
| FS | 39.6 [$\sigma$=2.1] | 18.6 [$\sigma$=0.3] |

Interestingly, QCEC and GS, which are the two prior methods that include data augmentation, are the only prior methods that substantially shrink the gap metrics (at the cost of balanced accuracy for $p < 0.9$).

Overall, this experiment shows that SPIRE is an effective mitigation method and that our evaluation framework enables us to easily understand how methods affect the behavior of a model. We use FS as the baseline for comparison for the remaining experiments because, of the prior methods, it had the best average balanced accuracy across $p$'s range.

## 5.2 Full Experiment

We evaluate SPIRE "in the wild" by identifying and mitigating SPs learned by a multi-label binary object-classification model trained on the full COCO dataset. Appendices D and G have additional details.

**Identification.** Out of all possible (Main, Spurious) pairs, we consider those which have at least 25 training images in Both ($\approx 2700$). From these, SPIRE identifies 29 where the model's prediction changes at least 40% of the time when we remove Spurious. Table 2 shows a few of the identified SPs; overall, they are quite diverse: the spurious object ranges from common (e.g., person) to rare (e.g., sheep); the SPs range from objects that are commonly co-located (e.g., tie-person) to usually separate (e.g., dog-sheep); and a few Main objects (e.g., tie and frisbee) have more than one associated SP. Notably, SPIRE identifies negative SPs (e.g., tie-cat) while prior work (Shetty et al., 2019; Singh et al., 2020; Teney et al., 2020) only found positive SPs. Appendix G includes a discussion of how we verified the veracity of these identified SPs.

**Mitigation.** Unlike the Benchmark Experiments, this experiment requires mitigating many SPs simultaneously. Because we found that SPIRE was more effective if we only re-train the final layer of the model in the Benchmark Experiments (see Appendix E), we do this by re-training the slice of the model's final layer that corresponds to Main's class on an augmented dataset that combines SPIRE's augmentation for each SP associated with Main. All results shown (Tables 3 and 4) are averaged across eight trials.

We conclude that SPIRE significantly reduces the model's reliance on these SPs based on two main observations. First, it increases balanced AP by 1.1% and shrinks the average recall/hallucination gaps by a factor of 14.2/28.1%, relative to the baseline model, on COCO. As expected, this does slightly decrease Mean Average Precision (MAP) by 0.4% on the original (biased) distribution. Second, it increases MAP on the UnRel (Peyre et al., 2017) and SpatialSense (Yang et al., 2019) datasets. Because this evaluation was done in a zero-shot manner and these datasets are designed to have objects in unusual contexts, this is further evidence that SPIRE improves distributional robustness. In contrast, FS decreases the model's performance, has inconsistent effects on the gap metrics, and has mixed results on the zero-shot evaluation.

**SPIRE and Distributional Robustness.** Noting that robustness to specific distribution shifts is one of the consequences of mitigating SPs, we can contextualize the impact of SPIRE by considering an extensive meta-analysis of methods that aim to provide general robustness (Taori et al., 2020). This analysis finds that the only methods that consistently work are those that re-train the baseline model on several orders of magnitude more data. It also describes two necessary conditions for a method to work. Notably, SPIRE satisfies both of those conditions: first, it improves performance on the shifted distributions (i.e., the balanced distributions, UnRel, and SpatialSense) and, second, this improvement cannot be explained by increased performance on the original distribution. Consequently, SPIRE 's results are significant because they show improved robustness without using orders of magnitude more training data. We hypothesize that SPIRE is successful because it targets specific SPs rather than using a less targeted approach.

## 5.3 Generalization Experiments

We illustrate how SPIRE generalizes beyond the setting from COCO, where we considered the object-classification task and assumed that the dataset has pixel-wise object-annotations. Specifically, we explore three examples that consider a different task (*Generalization 1*) or do not assume this (*Generalization 2*).

**Scene Identification Experiment (Generalization 1).** In this experiment, we construct a scene identification task using the image captions from COCO and show that SPIRE can identify and mitigate a naturally occurring SP. To do this, we define two classes: one where the word "runway" (the part of an airport where airplanes land) appears in the caption (1,134 training images) and another where "street" appears (12,543 training images); images with both or without either are discarded. For identification, we observe that removing all of the airplanes from an image of a runway changes the model's prediction 50.7% of the time.

Table 5 shows the results. SPIRE reduces the model's

Table 5: Results for the Scene Identification Experiment (averaged across sixteen trials).

| | Original AP | Balanced AP | %Δ Avg. Recall Gap | %Δ Avg. Hallucination Gap |
|---|---|---|---|---|
| Baseline | **95.0** | 48.9 | — | — |
| SPIRE | 92.8 [$\sigma$=1.6] | **83.2** [$\sigma$=7.3] | **-82.1** | **-75.5** |
| FS | 93.7 [$\sigma$=1.3] | 47.8 [$\sigma$=8.6] | -11.5 | 3.7 |

Table 6: Results for the No Object Annotation Experiment for the tennis racket example (averaged across eight trials).

| | Original AP | Balanced AP | %Δ Avg. Recall Gap | %Δ Avg. Hallucination Gap |
|---|---|---|---|---|
| Baseline | 93.9 | 79.9 | — | — |
| SPIRE | 92.9 [$\sigma$=0.4] | 80.5 [$\sigma$=1.3] | **-31.3** | -27.3 |
| SPIRE-R | **94.0** [$\sigma$=0.6] | **81.0** [$\sigma$=1.7] | -9.1 | **-44.9** |
| FS | 92.9 [$\sigma$=1.0] | 80.7 [$\sigma$=2.2] | -10.4 | -22.3 |

Table 7: Results for the ISIC Experiment (averaged across eight trials).

| | Original AP | Balanced AP | %Δ Avg. Recall Gap | %Δ Avg. Hallucination Gap |
|---|---|---|---|---|
| Baseline | 78.3 | 71.0 | — | — |
| SPIRE-EM | **78.8** [$\sigma$=4.9] | **76.4** [$\sigma$=2.4] | -20.5 | -39.0 |
| FS | 70.7 [$\sigma$=6.8] | 68.0 [$\sigma$=3.8] | **-31.3** | **-61.0** |

reliance on this SP because it substantially increases balanced AP and it reduces the average recall and hallucination gaps by factors of 82.1% and 75.5%. In contrast, FS is not effective at mitigating this SP.

**No Object Annotation Experiment (Generalization 2).** In this experiment, we mitigate the SP from the tennis racket example without assuming that we have pixel-wise object-annotations; instead, we make the weaker assumption that we have binary labels for the presence of Spurious. To do this, we train a linear (in the model's representation space) classifier to predict whether or not an image contains a person (similar to Kim et al. (2018)). Then, we project across this linear classifier to essentially add or remove a person from an image's *representation* (so we call this method SPIRE-R) (see Appendix H.1).

Table 6 shows the results. First, note that SPIRE provides a small increase in Balanced AP while providing the largest average decrease in the gap metrics. Second, note that SPIRE-R is preferable to FS because it produces a larger reduction in the hallucination gap while being otherwise comparable.

**ISIC Experiment (Generalizations 1 & 2).** In this experiment, we imitate the setup from (Rieger et al., 2020) for the ISIC dataset (Codella et al., 2019). Specifically: the task is to predict whether an image of a skin lesion is malignant or benign; the model learns to use a SP where it relies on a "brightly colored sticker" that is spuriously correlated with the label; and the dataset does not have annotations for those stickers to use create counterfactual images. For this experiment, we illustrate another approach for working with datasets that do not have pixel-wise object-annotations: using *external models* (so we call this method SPIRE-EM) to produce potentially noisy annotations with less effort than actually annotating the entire

dataset. The external model could be an off-the-shelf model (e.g., a model that locates text in an image) or a simple pipeline such as the super-pixel clustering one we use (see Appendix H.2).

Table 7 shows the results. We can see that SPIRE-EM is effective at mitigating this SP because it generally improves performance while also shrinking the gap metrics.[2] In contrast, FS does not seem to be beneficial because it substantially reduces performance on both the original and balanced distributions (which outweighs shrinking the gap metrics).

## 6 Conclusion

In this work, we introduced SPIRE as an end-to-end solution for addressing Spurious Patterns for image classifiers that are relying on spurious objects to make predictions. SPIRE identifies potential SPs by measuring how often the model's prediction changes when we remove the Spurious object from an image with a positive label and mitigates SPs by shifting the training distribution towards the balanced distribution while minimizing any correlations between the label and artifacts in the counterfactual images. We demonstrated that SPIRE is able identify and, at least partially, mitigate a diverse set of SPs by improving the model's performance on the balanced distribution and by making it more robust to specific distribution shifts. We found that these improvements lead to improved zero-shot generalization to challenging datasets. Then, we showed that SPIRE can be applied to tasks other than object-classification and we illustrated two potential ways to apply SPIRE to datasets without pixel-wise object-annotations to use to create counterfactuals (creating counterfactual representations and using external models). More generally, in terms of identifying SPs, our findings provide additional evidence for the claim that a correlation is neither sufficient nor necessary for a model to learn a SP.

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

## A    Discussion

In this section, we elaborate on SPIRE's strengths, its weaknesses, and suggested directions for future work. While there are many ways to improve SPIRE, we have demonstrated that it is a clear step forwards for the problem of addressing SPs.

**Generating Counterfactual Images.** SPIRE relies on the ability to produce counterfactual images. As a result, finding ways to produce similar counterfactuals with fewer assumptions (e.g., being able to add/remove objects without relying on having an annotated dataset) or to produce different types of counterfactuals (e.g., changing attributes such as "color") are both directions for future work. The former would improve the general applicability of SPIRE while the later would increase the scope of the types of SPs SPIRE could address.

**Identification.** SPIRE's strategy for identification can be summarized as "measure the probability that the model's prediction changes when we take an image from Group $X$ and apply Counterfactual Transformation $Y$." Intuitively, this strategy is effective because the original and counterfactual versions of an image differ only in terms of the effect of the counterfactual transformation while, if we were to compare natural images in one group to another group, there are probably going to be additional differences. Because SPIRE uses $X$ = Both, it may not be as effective as possible for identifying negative SPs because this split is likely to be very small for negatively correlated objects. As a result, future work could increase the scope of the types of SPs SPIRE could identify by considering different definitions of $X$ (e.g., $X$ is the set of images that have objects $1, \ldots, m$ and do not have objects $m + 1, \ldots, n$; $X$ is the set of images where objects 1 and 2 appear near to/far from each other) or $Y$ (e.g., $Y$ removes objects 1 and 2; $Y$ changes the location of object 1).

Interestingly, we find that a strong correlation is neither sufficient (Figure 7 shows that the model can ignore a strong correlation) nor necessary (Table 2 shows that some SPs are between objects that are almost uncorrelated) for a model to learn to use a SP, which is consistent with prior findings (Shah et al., 2020; Nagarajan et al., 2021). These result demonstrates SPIRE's advantage over identification methods that only consider the training distribution (e.g., Wang et al., 2020).

**Mitigation.** To begin with, it is worth noting that mitigating a SP may not always be worthwhile (e.g., when one is certain that the distribution will not shift).

At a high level, SPIRE's strategy for mitigation works by removing the statistical incentive for the model to rely on the SP, while trying not to add new SPs; this strategy may be less effective for SPs that do not arise from correlations in the training distribution. Previous augmentation-based mitigation methods might be less effective because they are intuitive rather than statistical (e.g., it makes intuitive sense that removing people should lessen the model's reliance on people to detect tennis rackets, but this intuition does not carry over to the dataset statistics). Previous regularization-based mitigation methods might be less effective because they may interfere with the learning process (e.g., cause the model to become stuck in a local minimum) or they may have effects that are too local to matter (e.g., changing the model's gradient at a point may not change its predictions very far away from that point). In particular, the Feature Splitting (FS) method from (Singh et al., 2020) assumes that one half of the features learned by the model are relevant for detecting objects "in context" and that the other half are relevant for objects "out of context;" while plausible for a single SP, this assumption becomes more tenuous as the number of SPs being mitigated increases.

While SPIRE's mitigation strategy is defined by two high-level goals, it is not always successful at realizing those goals (e.g., for $p < 0.5$ in Section 5.1, SPIRE introduces the potential for new SPs) and the way those goals are realized depends on the problem setting. As a result, future work could improve the general applicability of SPIRE by finding a unified strategy that does not depend on the problem setting, generalizing that strategy to work for more general SPs, and extending that strategy to problems other than image classification. Additionally, future work could develop a theoretical framework to help understand the effects of augmentation-based mitigation strategies.

# B  Method Details

## B.1  When can SPIRE be applied to a problem?

In this section, we walk through the assumptions needed to apply SPIRE to a *model* that has been trained to perform some *task* using some *data*.

- SPIRE is model-agnostic and, as a result, can be applied to any type of model.
- SPIRE assumes that task is binary-classification and, consequently, that each image has a binary label.
- SPIRE assumes that each image has a different binary label for the "spurious feature" that we are interested in studying. For example, this could be variable indicating whether or not the image contains some high frequency signal.
- SPIRE assumes we can manipulate the images in such a way that we can create a counterfactual version of an image in one split that belongs to a different split. This entails being able to change an image's label and/or the value of the "spurious feature."

Collectively, these assumptions allow us to define the image splits (Table 8) and produce the counterfactual images that SPIRE uses to identify and mitigate spurious patterns.

## B.2  Generating Counterfactual Images

Similar to prior work, SPIRE generates counterfactual images by adding objects to or removing objects from the original image (Shetty et al., 2019; Teney et al., 2020; Xiao et al., 2021; Chen et al., 2020a; Liang et al., 2020; Agarwal et al., 2020). In this work, use the pixel-wise object-annotations that are part of various datasets such as COCO to generate the counterfactual images. Figure 5 shows examples. Orthogonally, there is prior work that generates fundamentally different types of counterfactual images (Neto, 2020; Zhang & Sang, 2020; Goel et al., 2021; Sauer & Geiger, 2021).

**Removing an Object.** We consider two different ways to define which region of the image we are going to replace (pixel-wise or bounding-box) and two different ways to in-fill that region (using constant grey color or in-painting with the model from (Nazeri et al., 2019)). When we say that we "remove" an object, we mean that we found its bounding-box region and in-filled it with grey. When we say that we "in-paint" an object, we mean that we found its pixel-wise region and in-painted it. In order to minimize label noise, we make sure we do not include Main in the region that is going to be removed when we are removing Spurious.

**Adding an Object.** To add an object to an image, we find the pixel-wise region for that object in a different image and then replace that region's counterpart in the original image with it. In order to minimize label noise, we make sure that we do not cover Main when we add Spurious.

Table 8: A more general definition of the *image splits* defined in Section 3.

| Binary Image Label | Binary "Spurious Feature" | Image Split |
|---|---|---|
| 1 | 1 | Both |
| 1 | 0 | Just Main |
| 0 | 1 | Just Spurious |
| 0 | 0 | Neither |

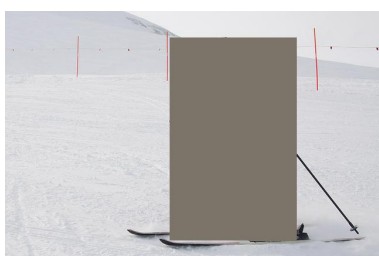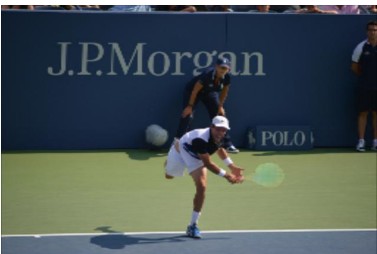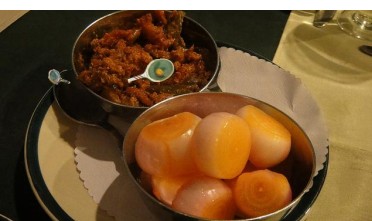

Figure 5: Example counterfactual images for the tennis racket example. **(Left)** An example of moving an image from Just Spurious to Neither by Removing Spurious. **(Center)** An example of moving an image from Both to Just Spurious by In-Painting Main. **(Right)** An example of moving an image from Neither to Just Main by Adding Main.

### B.3 What does it mean to introduce new potential SPs?

We try to minimize the potential for new SPs by ensuring that P(Main | Artifact) = 0.5, where the Artifact could be "Grey Box" from removing objects from an image or objects with "Unusual Placement" from adding objects to an image. However, it is not clear whether 0.5 or P(Main) is the "correct" choice for this value. One one hand, using P(Main | Artifact) = 0.5 maximizes the loss that the model will receive if it relies on Artifact. On the other hand, setting P(Main | Artifact) = P(Main) means that Main is independent of Artifact and that there is no statistical incentive for the model to rely on Artifact. Because we will not be evaluating the model (in terms of accuracy) on images with Artifact, we chose 0.5 because it actively discourages using Artifact rather than simply not encouraging it.

### B.4 Setting 1: Working through SPIRE's augmentation strategy

In this setting, {Both, Neither} each have size $0.5p$ while {Just Main, Just Spurious} have size $0.5(1-p)$.

For $p > 0.5$, SPIRE removes {Main, Spurious} from Both with probability $\frac{2p-1}{2p}$ and, as a result, P(Main | Grey Box) = 0.5. Similarly, SPIRE adds {Main, Spurious} to Neither with the same probability and, as a result, P(Main | Unusual Placement) = 0.5. As a result, {Just Main, Just Spurious} each receive $0.25(2p-1)$ images from each of {Both, Neither} and have an augmented size of $0.5p$. So SPIRE produces the balanced distribution without creating the potential for new SPs.

For $p < 0.5$, SPIRE adds Main to Just Spurious and adds Spurious to Just Main with probability $\frac{p-0.5}{p-1}$ and, as a result, P(Main | Unusual Placement) = 1. Similarly, SPIRE removes Spurious from Just Spurious and removes Main from Just Main with the same probability and, as a result, P(Main | Grey Box) = 0. As a result, {Both, Neither} each receive $0.5(0.5-p)$ images from each of {Just Main, Just Spurious} and have an augmented size of $0.5(1-p)$. So SPIRE produces the balanced distribution while creating the potential for new SPs.

## C   Evaluation Details

### C.1   Why not set P(Spurious | Main) = P(Spurious | not Main) = P(Spurious) for the Balanced Distribution?

Using P(Spurious) instead of 0.5 may be an intuitive choice because it would mean that the main statistical difference between the original and balanced distributions is that Main and Spurious are now independent. However, doing so can have dramatic and unexpected effects on which splits are more important for evaluation. To see this, consider Main = "fork" and Spurious = "dining table". For the original distribution, we have P(Spurious | Main) = 0.76 which means we have, roughly, a 3:1 ratio of images in Both to Just Main. For the balanced distribution, using $\lambda$ = P(Spurious) = 0.1 would change this ratio to 1:9. Not only would this choice change which split is more important for evaluation (from Both to Just Main) but it would also would increase the degree to which that split is more important (from a factor of 3 to a factor of 9). Without domain knowledge telling us that such a dramatic shift is warranted, using 0.5 is the more conservative option because assigning equal importance to images with and without Spurious never flips which splits are more important for evaluation.

### C.2   Why do small, in absolute terms, Hallucination gaps matter?

To understand this, consider the per split accuracies for the tennis racket example (Figure 3 Left) where we observe that the Hallucination gap is "only" 0.5% and may be tempted to conclude that it is not significant. However, when we look at where the model's errors come from on the original distribution, we find that roughly 40% of them come from Just Spurious, despite the model's 99.5% accuracy on this split. This means that the model's performance is sensitive to both small changes to its accuracy on Just Spurious and Neither and distribution shifts that move weight between Just Spurious and Neither.

As a result, small, in absolute terms, changes to the Hallucination gap can have large impacts on the model's robustness to distribution shift. In general, we adjust for this by measuring changes in the gap metrics relative to their original value (e.g., if the new model had a hallucination gap of 0.25% we would say that it "reduced the hallucination gap by a factor of 50%").

### C.3   Why can the Gap Metrics change much more than performance on the Balanced Distribution?

In general, mitigation methods shrink the gap metrics by sacrificing accuracy on the splits where relying on the SP is helpful in order to gain accuracy on the other splits; whether or not this trade-off improves performance on the balanced distribution depends on how much accuracy is sacrificed and gained. As a result, the size of the gap metrics and performance on the balanced distribution are not necessarily closely connected. As an extreme example of this, consider a hypothetical mitigation method that works by reducing the model's accuracy on the higher performing splits to match its accuracy on the lower performing splits: this will improve the gap metrics by setting them to zero, but it will harm performance on the balanced distribution.

### C.4   Counterfactual Evaluation

While the evaluations described in Section 4 are all based on the natural images, we also run a *counterfactual evaluation*. Unlike in SPIRE's identification step, where we only measure the probability that "removing Spurious from an image from Both" changes the model's prediction, this evaluation measures the probability that the model's prediction changes when we move an image from one split to another for each pairs of splits that differs by one object. This acts as an additional sanity check that a mitigation strategy has reduced the model's reliance on a SP, but we consider it to be less important than the model's performance on the balanced distribution and the gap metrics because its results depend on the specific definition of the counterfactuals used (e.g., it is easy to do well on this evaluation for a specific type of counterfactual by training the model on that same type of counterfactual).

# D  Tennis Racket Example: Metrics and Mitigation Results.

Here, we walk through the evaluation described in Section 4 for class imbalanced problems using the tennis racket example. Figure 6 shows the results. The numbers in the legends are "mean (standard deviation)" across 8 trials for the metric measured in that plot.

*Top Left: Average Precision.* This panel shows the model's Precision vs Recall curve, for the balanced distribution, which we use to calculate Average Precision by finding its Area Under the Curve (AUC). SPIRE improves Average Precision on the balanced distribution for this SP by 0.6%.

*Top Middle: Average Recall Gap.* This panel shows the model's recall gap (the absolute value of the difference of the model's accuracy on Both and Just Main) vs its Recall on the balanced distribution. We calculate this metric by finding the AUC. SPIRE decreases this metric by 31.4% which means that it produces a model that is more robust to distribution shifts that move probability between Both and Just Main.

*Top Right: Average Hallucination Gap.* This panel shows the model's hallucination gap (the absolute value of the difference of the model's accuracy on Just Spurious and Neither) vs its Recall on the balanced distribution. We calculate this metric by finding the AUC. SPIRE decreases this metric by 25.0% which means that it produces a model that is more robust to distribution shifts that move probability between Just Spurious and Neither.

*Center Row: Accuracy on Both and Just Main.* These panels plot the model's accuracy on Both/Just Main vs its Recall on the balanced distribution. The value shown is the AUC of this curve. Because the baseline model uses the presence of a person to help detect a tennis racket, we expect a model that does not rely on this SP to lose accuracy on Both and gain it on Just Main. SPIRE does this.

*Bottom Row: Accuracy on Just Spurious and Neither.* These panels plot the log of the model's accuracy on Just Spurious/Neither vs its Recall on the balanced distribution. The value shown is the AUC of this curve (before taking the log). Because the baseline model uses the presence of a person to help detect tennis rackets, we expect a model that does not rely on this SP to lose accuracy on Neither and gain it on Just Spurious. Because SPIRE improved AP, we do not see this because it's accuracy on these splits is higher for most levels of recall.

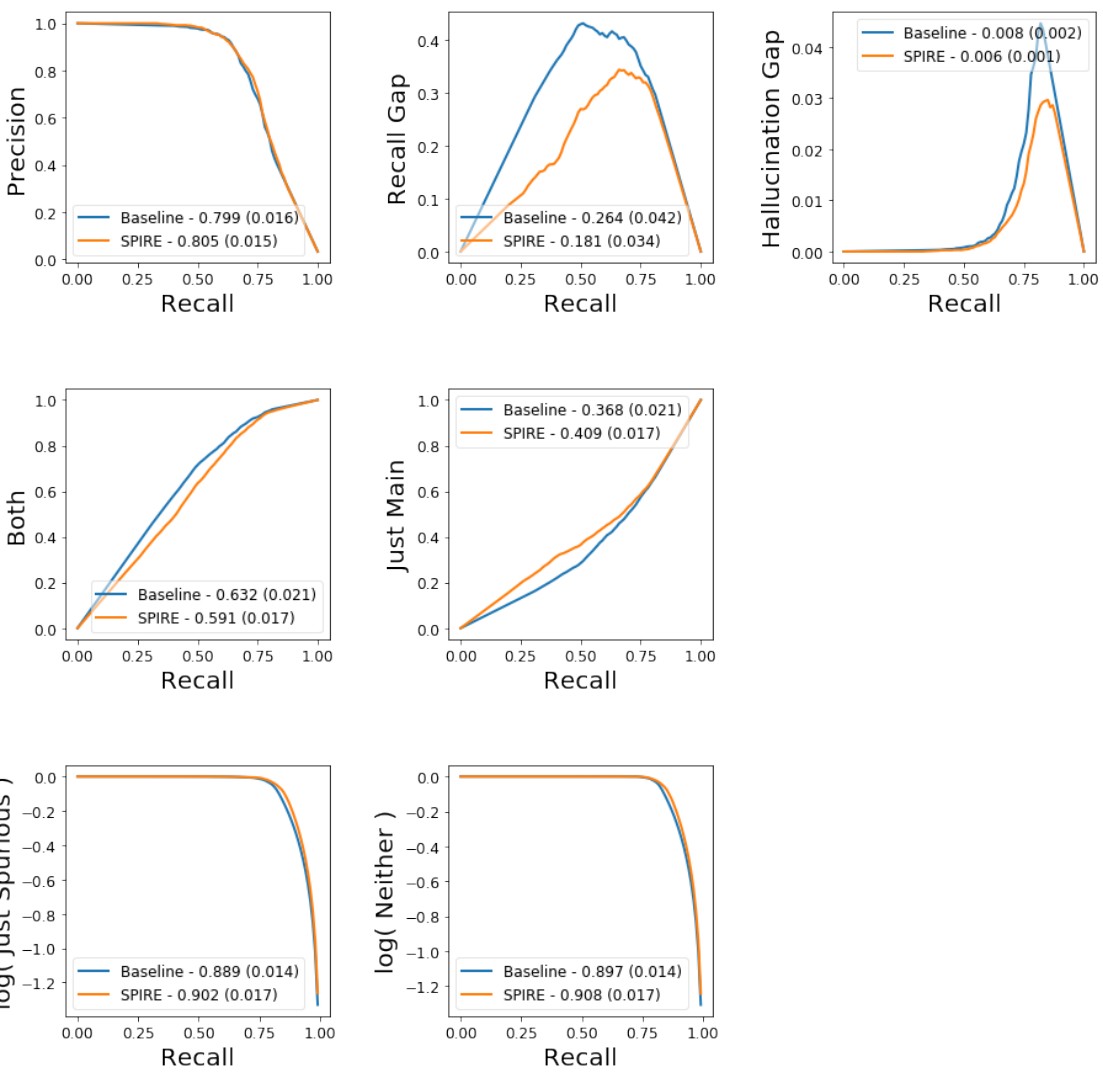

Figure 6: The results of our evaluation for the tennis racket example. The numbers in the legends are "mean (standard deviation)" across 8 trials. SPIRE improved Average Precision on the balanced distribution by 0.6%, decreased the average recall gap by 31.4%, and decreased the average hallucination gap by 25.0%. Further, it had the expected effect of decreasing accuracy on Both and increasing it on Just Main. As a result, we conclude that it reduced the model's reliance on this SP.

## E Model Training Details

Many of our experiments are based on the COCO dataset (Lin et al., 2014). Because the test set for this dataset is not publicly available, we used its validation set as our test set and divided its training set into 90-10 training and validation splits.

All of our experiments started with the pretrained ResNet18 (He et al., 2016) that is available from PyTorch (Paszke et al., 2019). For each task, the classification layer was replaced with one of the appropriate dimension and then trained via transfer learning (i.e., only the classification layer had its weights updated). The resulting model was then fine-tuned (i.e., all of its weights were updated) to produce what we call the Baseline Model throughout this work.

**Optimization.** We minimized the binary cross entropy loss using Adam (Kingma & Ba, 2014) with a batch size of 64. For transfer-learning, we used a learning rate of 0.001 and, for fine-tuning, we used a learning rate of 0.0001; we explored other options during early experiments, but found there was no benefit to doing so. If the training loss failed to decrease sufficiently after some number of epochs, we lowered the learning rate.

**Model selection.** During the training process, we selected the best model weights using their performance on the validation set. For the Benchmark Experiments, we measured performance using Accuracy and, for the Full Experiment, No Object Annotation Experiment, Scene Identification Experiment, and ISIC Experiment, we used F1. If the validation performance failed to increase sufficiently after some number of epochs, we stopped training.

**Benchmark Experiments: Hyper-parameter selection.** For this experiment, we tuned the hyper-parameters using balanced accuracy on the bottle-person pair with $p = 0.95$. For all methods, we considered both transfer-learning and fine-tuning, as applicable. For SPIRE, we considered both removing objects by covering them with a grey box and by in-painting them; we found that transfer-learning while covering objects with a grey box was the most effective (see Table 9). RRR, CDEP, and GS all have regularization weights that can be tuned. FS has a minimum weight for images of objects "out of context" that can be tuned. For these methods, we considered values that are powers of 10 ranging from 0.1 to 10,000; no method chose one of the extreme values.

**Full Experiment: Hyper-parameter selection.** For this experiment, we tuned the hyper-parameters using the mean, across SPs, Average Precision on the balanced distribution for a model trained on 50% of the training dataset and then evaluated on the remaining 50% of the training dataset; we used such large chunk of the dataset for evaluation in order to be able to estimate the per split accuracies, which are required to calculate Average Precision on the balanced distribution.

For SPIRE, we use transfer-learning while covering objects with a grey box because this is what we found worked best in the Benchmark Experiments. However, we tune the weight of the augmentation by scaling $\delta$ from Setting 2 in Section 3.1 by a factor of $\{0.1, 0.2, 0.3, 0.4, 0.5, 0.6, 0.7, 0.8, 0.9, 1.0\}$; intuitively, this is to prevent us from adding too many counterfactual images. Note that the weight for each SP is tuned independently and each weight is tuned by training a linear classifier.

For FS, the configuration chosen by this procedure yielded poor results and, consequently, we used the default value of 3 for our results (Singh et al., 2020).

Table 9: The results of the Hyper-parameter Selection for SPIRE on the Benchmark Experiments. We see that SPIRE is consistently more effective when it retrains the model using transfer-learning and when it removes objects by covering them with grey boxes. Results shown are averaged across eight trials.

| Removal Strategy | Parameters Adjusted | Mean (Standard Deviation) |
|---|---|---|
| Grey Box | Transfer-Learning | 70.4 (1.7) |
| | Fine-Tuning | 69.4 (1.3) |
| In-Painting | Transfer-Learning | 70.2 (1.3) |
| | Fine-Tuning | 68.2 (1.4) |

# F   Additional Results: Benchmark Experiments - Section 5.1

**Creating the benchmark.** While varying $p$ allows us to control the strength of the correlation between Main and Spurious (i.e., $p$ near 0 indicates a strong negative correlation while $p$ near 1 indicates a strong positive correlation), it does not guarantee that the model actually relies on the intended SP. Indeed, when we plot the models' balanced accuracy as $p$ varies (Figure 7), we observe that 5 out of the 13 pairs show little to no loss in balanced accuracy as $p$ approaches 1 (dashed lines). Consequently, subsequent evaluation considers the other 8 pairs (solid lines). For these pairs, the model's reliance on the SP increases as $p$ approaches 0 or 1 as evidenced by the increasing loss of balanced accuracy and confirmed via counterfactuals (Figure 8).

**Counterfactual Evaluation.** For models that are trained on a dataset augmented with a specific type of counterfactual images, the results of this evaluation for that type of counterfactual are often skewed and, consequently, we exclude those results. Specifically, this means that: SPIRE is only evaluated on In-Painting counterfactuals, QCEC is not evaluated on In-Painting counterfactuals, and GS is not evaluated on counterfactuals that In-Paint Main.

Figure 9 shows the results (averaged across the chosen object pairs and eight trials per pair). The first thing to note is that all of the counterfactual evaluations show that the Baseline model is relying on the intended SP because their results get worse as P(Main | Spurious) approaches 0 or 1 (i.e., there is a strong negative or positive correlation between Main and Spurious in the training dataset). Observe that SPIRE improves all of evaluations based on In-Painting with the exception of "Just Spurious and In-Paint Spurious" for $0.05 < p < 0.5$. In contrast, the other mitigation methods have clear and consistent failures (e.g., RRR, CDEP, GS, and FS all make the evaluation worse for "Both and Remove Main", QCEC makes the evaluation worse for "Neither and Add Main").

**Per split analysis.** By looking at the models' accuracy on each split (Figure 10, averaged across the chosen object pairs and eight trials per pair), we see that SPIRE exhibits all of the expected signs of a method that is reducing a model's reliance on a SP:

- It sacrifices accuracy on splits where relying on the SP is helpful (e.g., Both for $p > 0.5$ and Just Main for $p < 0.5$) in order to gain accuracy on the splits where the SP is not helpful (e.g., Just Main for $p > 0.5$ and Both for $p < 0.5$).
- It substantially flattens the per split accuracy curves for images with Spurious and, to a lesser extent, flattens them for images without Spurious. This indicates that it produces a model that is less sensitive to the original training distribution.

**Comparing Augmentation Strategies.** In this experiment, we want to explore the specific effect of SPIRE's augmentation strategy. We do this by comparing SPIRE to modified versions of the two augmentation-based methods that we consider:

- QCEC-Aug, which augments the dataset with images where either Main or Spurious has been removed (uniformly at random, as applicable). Compared to QCEC, it differs in that it removes objects by covering them with a grey box (as SPIRE does) instead of in-painting them.
- GS-Aug, which augments the dataset with images where Main has been removed (if possible). Compared to GS, it differs in that it it removes objects by covering them with a grey box (as SPIRE does) instead of in-painting them and in that it does not include GS's regularization term.

In Figure 11 shows the results:

- SPIRE is the most effective method for increasing Balanced Accuracy and decreasing the Recall Gap.
- While QCEC-Aug and GS-Aug are somewhat better at reducing the Hallucination Gap, this usually is not worth the loss in Balanced Accuracy.
- Note that the results for the baseline model and for SPIRE are slightly different than they are in Figure 4 because these results are based on re-running this experiment from scratch.

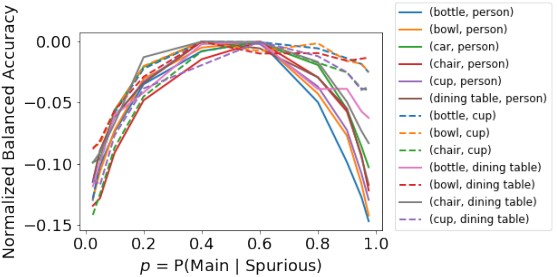

Figure 7: For each pair of objects, we plot the models' balanced accuracy as we vary $p$ for the training set. The y-axis is normalized so that we can easily compare the curvature of the plots. We either accept (solid line) or reject (dashed line) pairs based on whether or not we see a significant drop in balanced accuracy both as $p$ approaches 0 and as it approaches 1. The rejected pairs show an insufficient drop as $p$ approached 1.

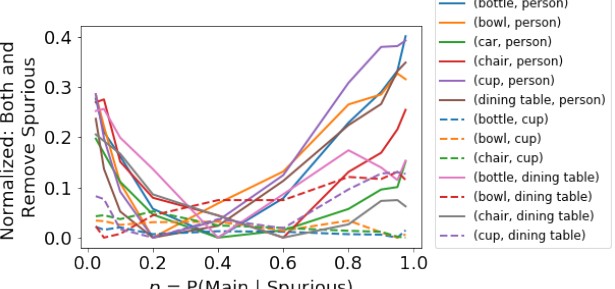

Figure 8: For each pair of objects, we plot the metric SPIRE uses to identify SPs as we vary $p$ for the training set. The y-axis is normalized so that we can easily compare the curvature of the plots. We see that SPIRE generally agrees with the decision to accept (solid line) or (dashed line) pairs from the benchmark because the accepted pairs generally show more curvature.

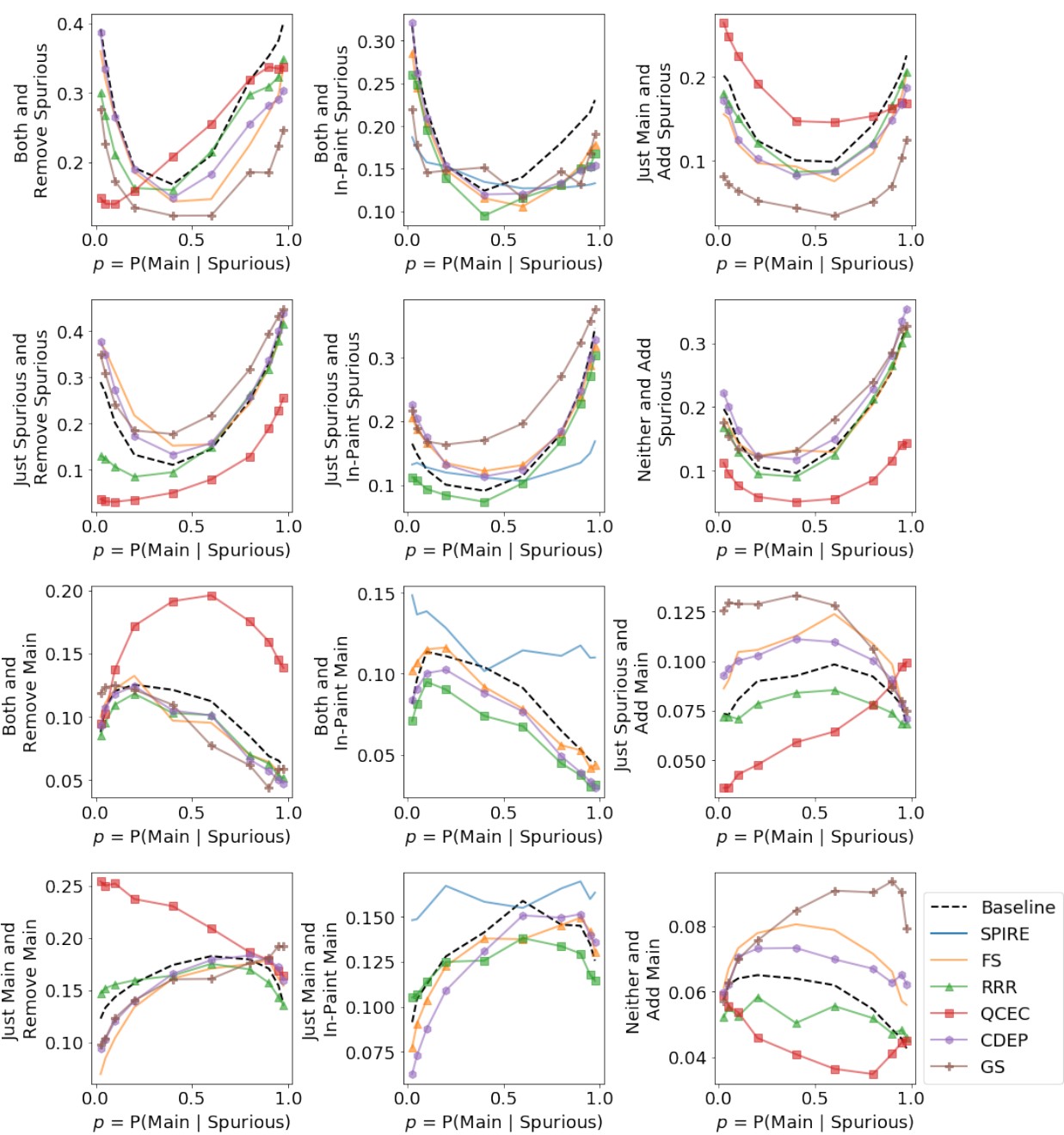

Figure 9: The columns correspond to Removing, In-Painting, and Adding an object. The first two rows do that to Spurious and, as a result, a lower value is better. The last two rows do that to Main and, as a result, a higher value is better. Methods that train on an augmented dataset that contains a certain type of counterfactual are excluded from its evaluation because their results are usually skewed.

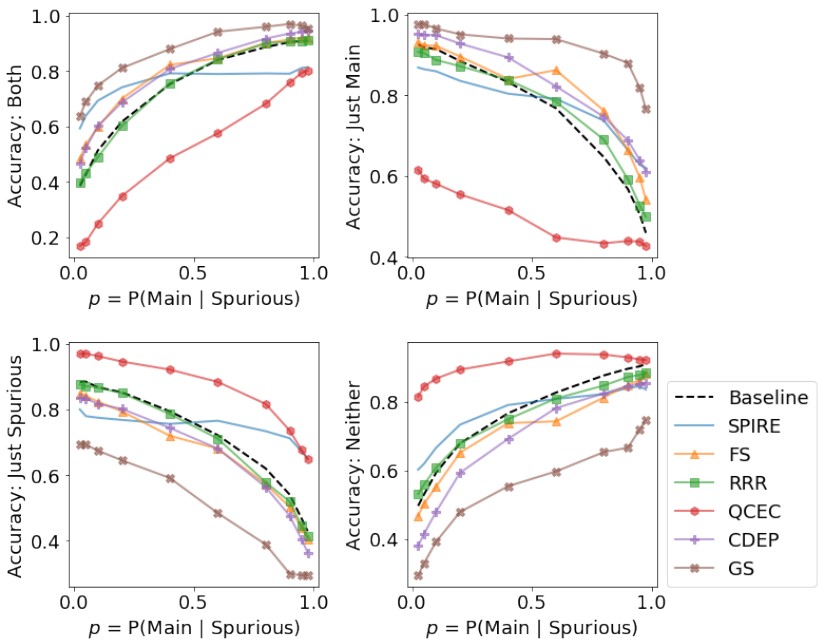

Figure 10: The models' accuracies on each split.

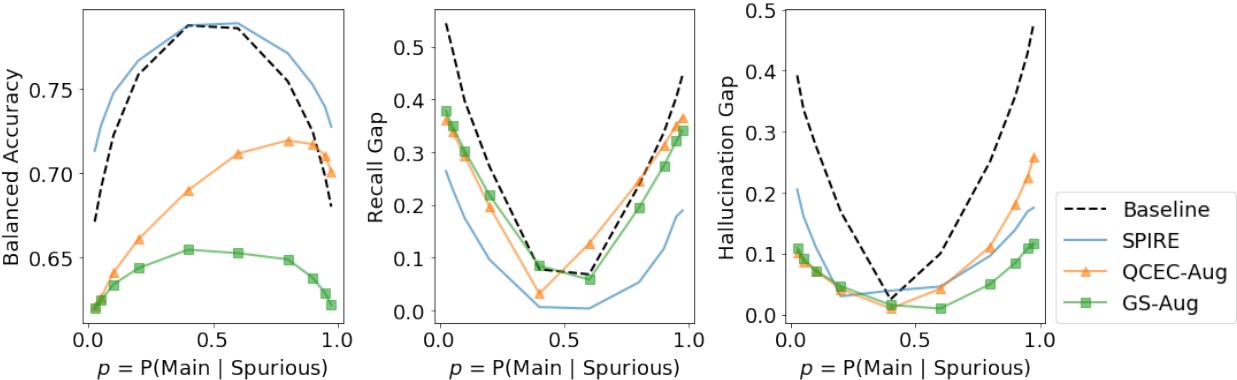

Figure 11: A summary of the comparison of various augmentation strategies for the Benchmark Experiments.

## G   Additional Results: Full Experiment - Section 5.2

**Validating the Identified SPs.** In Figure 12, we verify that the model is generally robust to masking objects. In Figure 13, we verify that the model has large recall and hallucination gaps for the identified SPs. Both of these results indicate, in different ways, that the model is indeed relying on the SPs that SPIRE identifies.

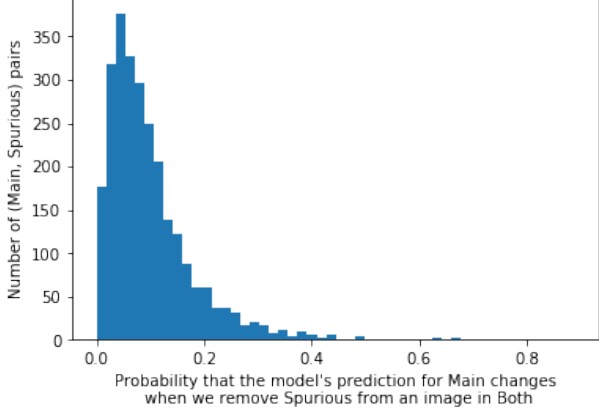

Figure 12: A histogram of the metric that SPIRE uses to identify SPs. In general, the model's predictions are quite robust to masking objects: changing a mean of 10.0% and a median of 7.7% of the time. As a result, it is unlikely we can explain the fact that the model's prediction changes more than 40% of the time for the identified SPs using the fact that these images are out-of-distribution (they contain grey boxes).

Figure 13:   When a model is relying on a SP, we expect positive gap metrics, if it has positive bias, and negative gap metrics, if it has negative bias. In general, this is what we find. **(Left)** A comparison of the Recall Gap to the bias of the dataset for the SP. **(Right)** A comparison of the Hallucination Gap to the bias.

**SPIRE's effect on each SP.** Figures 14, 15, and 16 show SPIRE's effect on the Balanced Average Precision, the Average Recall Gap, and Average Hallucination Gap respectively. SPIRE improved Balanced Average Precision by an average of 1.1% with a positive change for 21 of the SPs. SPIRE decreased the Average Recall/Hallucination Gaps by an average factor of 14.2%/28.1% for 24/29 of the SPs. Overall, these results indicate that SPIRE consistently reduces the model's reliance the identified SPs.

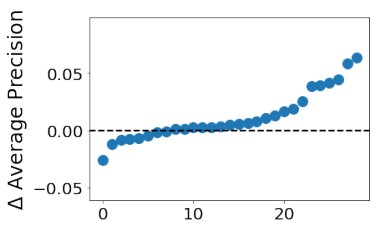 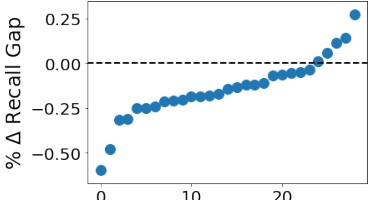 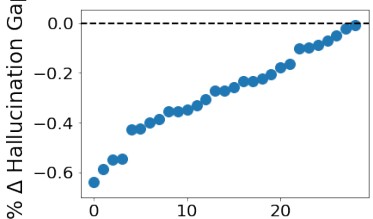

Figure 14: The Average Precision on the balanced distribution for SPIRE compared to the baseline for each SP.

Figure 15: The percent change of the Average Recall Gap for SPIRE compared to the baseline model for each SP.

Figure 16: The percent change of the Average Hallucination Gap for SPIRE compared to the baseline model for each SP.

# H    Additional Results: Generalization Experiments - Section 5.3

## H.1    SPIRE-R - Projection PseudoCode

Algorithm 1 details the process that we use for "adding" or "removing" Spurious from a model's representation. This process is guaranteed to change the initial linear model's prediction for whether or not the representation contains Spurious, which suites our goal of changing the most "obvious" signal for Spurious in the representation. Future work could explore more ambitious goals (e.g., removing all of the information about Spurious from the model's representation).

---

**Algorithm 1** Our algorithm for "adding" or "removing" Spurious from a model's representation.

---

**Require:** $\{r_i\}_{i=1}^n$            ▷ The model's penultimate layer's representation of each image
**Require:** $\{y_i\}_{i=1}^n$            ▷ A binary label for whether or not each image contains Spurious
**Require:** $c$            ▷ A confidence threshold, we used 0.0001
**Require:** $s$            ▷ A step size, we used 0.1
  $w, b \leftarrow LogisticRegression(\{r_i\}, \{y_i\})$    ▷ Train a linear model to predict if an image contains Spurious
  **for** $i = 1, \ldots, n$ **do**
    $r'_i = r_i$            ▷ Initialize the counterfactual representation
    **if** $y_i == 1$ **then**            ▷ If the image contains Spurious, remove it
      $y'_i \leftarrow 0$
      **while** $\frac{1}{1+e^{-(wr'_i+b)}} > c$ **do**
        $r'_i \leftarrow r'_i - s * w$    ▷ Maximally decrease the linear model's confidence that Spurious is present
      **end while**
    **else**            ▷ If the image doesn't contain Spurious, add it
      $y'_i \leftarrow 1$
      **while** $\frac{1}{1+e^{-(wr'_i+b)}} < 1 - c$ **do**
        $r'_i \leftarrow r'_i + s * w$    ▷ Maximally increase the linear model's confidence that Spurious is present
      **end while**
    **end if**
  **end for**
  **return** $\{r'_i\}, \{y'_i\}$

---

## H.2    ISIC Experiment- Pipeline for creating counterfactuals

Our pipeline, which is based on clustering image segments (i.e., super-pixels), is constructed as follows:

- We use an image segmentation algorithm to extract segments from an image and represent each segment using its mean RGB value.
- We run a hierarchical clustering on those RGB values to produce nine clusters. Then, we manually inspect several randomly sampled images from each cluster and label those clusters based on whether or not they represent stickers.
- Finally, we use a K-NearestNeighbor classifier to predict which of those nine clusters an image segment belongs to.

Overall, this pipeline produces a per-image map of which pixels belong to a sticker and identifies stickers with 86.7% recall and 99.0% precision. We use this map to produce counterfactual images.

Note that this pipeline significantly reduced the cost of producing these pixel-wise annotations because it only required labeling a small number of image segments as "sticker or not." However, the annotations are noisier than manually collected annotations would be.

