# OpenReview forum: "Finding and Fixing Spurious Patterns with Explanations"
_TMLR — Accepted by TMLR_

### Review · Reviewer_3jVt · 2022-06-17

**Summary Of Contributions:**

Image classification models often learn spurious correlations because of biases in the training dataset. For example, a model tends to correlate tennis rackets with people and therefore classifies tennis rackets only when people are present and not otherwise. Such behavior is referred to as spurious pattern (SP).

This paper presents a method, SPIRE, for identifying and mitigating SPs in image classification models. SPIRE, in its default state, requires bounding box annotations for all objects of the concerned classes in the training set. The paper proposes a couple of methods for working around this requirement, towards the end.

Identification of SPs: For each pair of class labels, SPIRE computes how often the removal of label-2 changes the prediction of label-1. This percentage can be thresholded to decide whether the pair of class labels presents an SP or not.

Mitigation: SPIRE manipulates the data by replacing objects with gray boxes or adding them into new images to create a balanced training set. SPIRE also takes care to not introduce an SP w.r.t. grey boxes. Either the model, or just the last layer, are then retrained on this balanced training set to mitigate the SP.

Evaluation: The authors report classification AP on the original dataset, AP on the balanced dataset (but in most cases not evaluating on manipulated images), and recall / hallucination gaps. Higher the AP on the balanced dataset, the better is the effect of mitigation; similarly lower the recall / hallucination gap, better is the mitigation.

More experiments: The authors demonstrate that after treating a model using SPIRE, it does better zero-shot generalization to UnRel and Spatial Sense datasets. They also show that SPIRE can be adapted to either SPIRE-R or SPIRE-EM. The former works without pixel level object annotations but uses image level annotations. The latter learns a few-shot nearest neighbor classifier on a handful of manually annotated images to get around the lack of annotations in the dataset to try to make SPIRE even more generally applicable when no formal image level annotations are available for the SP in question.

**Broader Impact Concerns:**

I do not see any broader impact concerns. Rather this work helps address the broader impact concerns of image classification models.

**Requested Changes:**

### Experiments

- For a handful of SPs, could Table 3 experiments be run with end-to-end fine-tuning? If I understood correctly, the problem with fine-tuning end-to-end in general is that one cannot mitigate multiple SPs simultaneously when doing so. But reporting results for a single SP at a time in this setting will be helpful. [Critical]

- Analysis: Demonstrate that identification reports no SPs when a similarly sized box not covering any object is grayed out. [Critical]

- Ablation: What if we gray out a random box in the image instead of the box around the correct object? This baseline should ideally demonstrate no mitigation or lower mitigation than SPIRE. [Critical]

### Presentation
- Please state explicitly the assumptions made for the type of annotations used by SPIRE for mitigation and identification. [Critical]
- In the submission, "Object detection" is often used to refer to the task of classifying whether an object is present in the scene or not. In computer vision, however, object detection refers to joint classification and localization of a tight fitting bounding box around the object. Please replace "object detection" with "object classification" or similar. [Critical]
- Figure 4 cannot be parsed without colour. Please use -. -* and other line styles to make this easier for colour blind people and for grayscale printing. [Moderate]
- Please provide equations for the projection in "generalization 2" experiments. [Critical]
- Please provide equations for average recall gap and hallucination gap (or pseudo code). [Moderate]
- Please rewise the claim "no dataset annotations" as manually inspecting several randomly sampled images from each cluster is data annotation. [Critical]
- Lots of repetitions were used in the experiments to report average performance but no error bars were reported. Please report error bars in atleast a few cases. [Critical]

### Typos
Page 6 bottom "Indeed, when -we- measure ..."



**Strengths And Weaknesses:**

### Strengths

SPIRE is a simple method and the paper makes simple claims and verifies them using experiments. The paper is of interest to a subset of the TMLR readership.

In particular, SPs are biases in the model which can be harmful depending on the application domain. They also limit out-of-distribution performance for instance when the spurious class is not present in the new domain or if two objects coexist in the new distribution but did not co-exist in the training set. SPIRE demonstrates that it mitigates SPs better than prior work.

Figure 4 is especially informative where the training dataset bias is varied and mitigation strategies are compared. SPIRE does not loose out on model performance while reducing bias.

### Weaknesses

Outside of issues in the presentation, which are discussed in the subsequent section, the main weakness is that most experiments use object level annotations which are often not available. SPIRE-R uses image level annotations but it is evaluated only on one SP. SPIRE-EM requires "manually inspect several randomly sampled images from each cluster" which is basically doing few-shot active learning to circumvent the lack of labels. So this is not "working without annotations".

During identification of SPs, how do you control for classification changes because of the input image being OOD due to the gray box? If the gray box did not cover either spurious or main, would the model output remain unchanged?

Another important weakness is that the experiments in Table 3 only fine-tune the last layer. What happens if we fine-tune end-to-end? Does it get worse because low level features adjust to gray boxes? Are we missing out by assuming that SP related issues could be fixed by changing the last layer only?

Lastly one ablation is missing: What if we gray out a random box in the image instead of the box around the correct object?

---

> ### Author Response · Authors · 2022-07-03
> **Main Response**
>
> **“The main weakness is that most experiments use object level annotations which are often not available”**
> We agree that this is a limitation of our work.  However, we would like to note that this is a fairly common assumption, as evidenced by the fact that all but one of the methods we compare against also rely on this assumption.  Moreover, we think that an interesting future direction would be to generalize these ideas to settings where such annotations are not available, and provide some preliminary evidence of the feasibility of this direction.
>
> **“For a handful of SPs, could Table 3 experiments be run with end-to-end fine-tuning? If I understood correctly, the problem with fine-tuning end-to-end in general is that one cannot mitigate multiple SPs simultaneously when doing so.”**
> We would like to clarify a potential misunderstanding: mitigating multiple SPs simultaneously is not incompatible with end-to-end fine-tuning:
> -  For example,  the FS results are based on end-to-end fine-tuning.
> -  SPIRE only fine-tunes the last layer because we found that to be more effective in the Benchmark Experiments.
>
> If the reviewer could point to the section of the paper that led them to this conclusion, that would be greatly appreciated and we will attempt to clarify it.
>
> **“Demonstrate that identification reports no SPs when a similarly sized box not covering any object is grayed out.”**
> Thank you for the suggestion.  While running this exact experiment is challenging (the COCO images are quite dense, so it isn’t always possible to add similarly sized gray boxes that don’t cover any object), we have added results for a similar experiment to the “Validating the Identified SPs” paragraph of Appendix G.
>
> Specifically, we have added a histogram of the metric SPIRE uses to identify SPs (the probability the model’s prediction for one object changes when we remove a different object from an image with both objects).  On the median across object pairs, the model's prediction changes 7.7% of the time.  So the model  is relatively robust to masking random objects and the OOD nature of these counterfactual images.  This is further evidence that the model is indeed relying on the identified SPs (where the prediction changes >40% of the time).
>
> **“What if we gray out a random box in the image instead of the box around the correct object?”**
> We believe the reason that you are suggesting that we run this ablation is to check if simply adding gray boxes to images results in more robust models (please let us know if we are misunderstanding).  If so, there are two things to note:
> -  As noted in the previous response, we observed that the model is already relatively robust to masking random objects.
> -  We ran modified versions of the QCEC and GS baselines that use the same counterfactuals as SPIRE.  As a result, the differences in the results between these baselines and SPIRE are due to the way in which SPIRE uses those counterfactuals.  The fact that SPIRE significantly outperforms these methods suggests that simply adding gray boxes is not enough.  We have added a discussion of this to the “Comparing Augmentation Strategies” paragraph of Appendix F
>
> **Clarify SPIRE’s Assumptions (“state explicitly the assumptions made for the type of annotations used by SPIRE for mitigation and identification”)**
> This was a common suggestion and one that we have acted upon in three ways:
> -  Throughout the paper, we have emphasized that SPIRE assumes that we have “access to pixel-wise object-annotations” for our specific experiments.
> -  At the end of the “Preliminaries” paragraph in Section 3, we have summarized the two general assumptions that must be satisfied in order to use SPIRE.
> -  We have added Appendix B.1, which goes through the process defining what types of problems SPIRE can be applied to and how to do so.
>
> **“Please revise the claim ‘no dataset annotations’ as manually inspecting several randomly sampled images from each cluster is data annotation”**
> Thank you for catching this.  We clarified that the goal of SPIRE-EM is to “produce potentially noisy annotations with less effort than actually annotating the entire dataset.”
>
> **“Please report error bars in at least a few cases.”**.
> We added a discussion on “Measuring Variance Across Trials” to Section 5 and then added the standard deviation of the change in Average Precision between the mitigated and the baseline model to Tables 3-7.

---

> > ### Comment · Reviewer_3jVt · 2022-07-04
> > **Follow-up**
> >
> > Thank you for addressing my questions and concerns.
> >
> > **Regarding final layer fine-tuning vs end-to-end fine-tuning**: I misunderstood based on these two lines. "Mitigation. Unlike the Benchmark Experiments, this experiment requires mitigating many SPs simultaneously. We do this by re-training the slice of the model’s final layer that corresponds to Main’s class on an augmented dataset that combines SPIRE’s augmentation for each SP associated with Main." in page 8. Please reword these lines and report end-to-end fine-tuning results for Table 3 alongside the current numbers. Practitioners using SPIRE would benefit from understanding the trade-offs involved there.

---

> > > ### Author Response · Authors · 2022-07-07
> > > **Thank you for clarifying**
> > >
> > > We agree that it makes sense to better explain why SPIRE focused on retraining only the last layer.
> > >
> > > For SPIRE, the choice of “which model parameters will be retrained” is a hyperparameter because it can be chosen without, implementation wise, influencing “how the dataset is augmented with counterfactual images”.  As a result, SPIRE only retrains the last layer because that is what worked best in the hyperparameter selection for the Benchmark Experiments.  We have clarified this in the lines that caused the reviewer’s confusion and have added the results of this hyperparameter selection to Table 9 of Appendix E.
> > >
> > > Relatedly, note that we used the results from the Benchmark Experiments to help design our subsequent experiments (including both the hyperparameter selection process for SPIRE and the choice of competitors to evaluate). As a result, we focused exclusively on only retraining the last layer for SPIRE in our subsequent experiments as well (including the results in Table 3).

---

> ### Author Response · Authors · 2022-07-03
> **Additional Response**
>
> The following additional concerns have been addressed;  thank you for your suggestions:
> -  **“Please replace ‘object detection’ with ‘object classification”**
> -  **“Figure 4 cannot be parsed without colour”**
> -  **“provide equations for the projection in ‘generalization 2’ experiments.”** Pseudo-code has been added to Appendix H.1
> -  **“provide equations for average recall gap and hallucination gap.”**  A reference to Appendix D, which provides details of how these metrics are calculated, has been added to Section 4.
> -  **“Typos”**

---

### Review · Reviewer_aW1B · 2022-06-27

**Summary Of Contributions:**

This paper presents SPIRE an approach for finding and fixing a model's dependence on spurious training artifacts. The approach is based on the assumption that several versions of training/validation dataset can be created such that objects in a scene/image can be partitioned into a 'main' object, 'spurious' object, 'both', and neither. If a model is relying on a spurious artifact, then its prediction changes when the spurious signal is not present in the image (neither, and main distributions).  Accuracy/performance gaps across the different data splits is used to detect whether a model is relying on a spurious signal.

In the second part of the paper, the work proposes to fix a model's dependence on spurious signals using counterfactual image generation. Specifically, the hypothesis here is that a model's dependence on spurious signals comes statistical skews in the data, i.e., all images with tennis rackets also include a person, so a person is often a spurious signal for tennis racket. To counter this, the proposed approach is via data augmentation, where a 'balanced' dataset is created such that the distribution of spurious and main signals are equalized; hence, the statistical skew is removed. Consequently, a model fined-tuned on this dataset has no use for the spurious signal. The paper considers two settings depending on whether there is class imbalance and it is possible to generate image counterfactuals where an object can be removed.

SPIRE is compared to several approaches that aim to reduce a model's dependence on spurious artifacts with favorable performance.

**Broader Impact Concerns:**

I don't see any ethical concern with this work.

**Requested Changes:**

I list the requested changes below. All of these changes are based on the weaknesses as discussed in the previous section. Overall, I don't see any of the following requested changes as disqualifying, but would help improve readability and impact of the work.

- **Writing/Clarity**: As noted in the previous section, Section 3.1 could use another pass to help improve clarity. I'd also implore the authors to explicitly clarify and state SPIRE assumptions and requirements upfront.

- **Dataset partition**: In the previous section, I list a bunch of questions relating to the dataset partition requirement. Some of these are partly addressed in the paper, but I believe that the requirement for a partition is the key to getting SPIRE to work. It'll be really helpful if these issues were further clarified in the paper.

- **The No Object Annotation Experiment**: This section needs further clarification. It is unclear what, "we project the linear classifier to essentially add or remove a person from the image" means. How was this done? How was this verified, i.e., is it still possible to predict presence of a person from the representations?

The questions above are the key ones that need to be further clarified in the draft.

**Strengths And Weaknesses:**

## Strengths

-  **Important Problem**: The task of identifying which spurious signal a model depends on, and then removing this reliance is an important one. This paper takes a stab at this problem for a concrete case/setting.

- **Simple and Straightforward Solution**: The augmentation strategy proposed is simple and feasible under controlled settings where the inputs can be partitioned into 'main', 'spurious' etc as required by this method. The idea of balancing the spurious and main signals in the new dataset is also particularly straightforward, and is shown to be effective for removing dependence on spurious signals.

- **Comprehensive Evaluation & Comparison with other methods**: Spurious signal detection is still a nascent problem. Several methods that claim to address this problem don't explicitly state their underlying assumptions, so it can be difficult to assess their effectiveness. This paper compares SPIRE to several proposed methods, so it is one of the first papers I have seen to perform a comprehensive evaluation, which is helpful for the field.

## Weaknesses
While this paper tackles an important problem, there are several weaknesses that'll highlight here. Some of these weakness are structural and probably unlikely to be easily fixed without overhauling the method, so I don't penalize the paper too much for those. In the requested changes section, I'll note the specific ones that should be addressed.

- **Clarity/Difficulty in Writing**: It is a bit difficult to digest portions of this paper, particularly section 3.1, which discuss the approach to creating the balanced datasets. In addition, SPIRE as a method/approach is not really properly spelled out. Here is what I mean: let's say I have new model trained on a different dataset than COCO or the ones discussed here, it is unclear what the specific assumptions that the training data, model, and task should meet for SPIRE to be useful.

- **Spurious Pattern Definition**: The definition of spurious signals in the opening paragraph is too vague. One could argue that any signal in the training set will satisfy this definition. I think the definition that the paper actually goes with is something like: a predictive signal that is correlated to the label for a particular distribution, but whose correlation 'goes away' once this distribution changes.

- **Data set partition question. Where do the 'main', 'just spurious', 'neither' etc partitions come from?**: It is unclear to me how one determines the sets main, spurious etc that is required for the SPIRE method to work. In the COCO dataset, I guess this is just assumed to be the possible object segmentations? In my opinion, defining these partitions or something akin to the partition is crucial for this approach to work. The No Object Annotation Experiment is interesting as a counter to my claim, however, I don't understand the specifics of this experiment enough. What does the statement: "we project across this linear classifier to essentially add or remove a person from an image's representation" mean? If I know the concept of a person already,  then you already have an implicit partitioning, so the process of removing the person from the representations is really the same as creating the just main distribution as you do in the COCO dataset. To me, this experiment doesn't quite demonstrate that the explicit pixel splits are not needed. I am fine with stating that SPIRE requires explicit pixel annotations to work, which can be a stated limitation of the method. In addition, why is tennis racket main and the person spurious? If the task were person detection, wouldn't these two categories be flipped? Further, does this mean that the data splits are dependent on the target signal/label being considered?

- **An example to help clear up my confusion**: Let's say we trained a COCO object detection model that happens to rely on some high frequency signals in the image for a particular object. How would the proposed partitions capture this signal? Note that this signal is not one that you have explicit pixel annotation for.

- **Spurious Signals from Memorized Examples**:  The proposed approached will catch spurious signals that are due to statistical skews (in the language of Nagarajan et al. (cited in the paper). However, recent work has suggested that it is possible for a model to learn a spurious signal from just *3* examples (see: https://arxiv.org/abs/2202.05189). This is likely out of scope for this work, but are the partitions required by this paper effective when a spurious dependence is induced by only a few examples.

---

> ### Author Response · Authors · 2022-07-03
> **Initial Response**
>
> **Clarify SPIRE’s Assumptions (“explicitly clarify and state SPIRE assumptions and requirements upfront” & “specific assumptions that the training data, model, and task should meet for SPIRE to be useful” & “Dataset partition”)**
> The reviewer had important questions about what SPIRE’s assumptions are and how the dataset partitions (ie, image splits) are defined.  These are related questions, so we will discuss how we revised the paper to address them simultaneously.
>
> First, we added a summary of the two general assumptions that SPIRE makes to the end of the “Preliminaries” paragraph in Section 3.
> -  The weaker assumption is being able to map an image to its appropriate image split.  As the reviewer noted, SPIRE-R still needs this assumption.
> -  The stronger assumption is being able to use counterfactuals to create a copy of an image from one split that belongs to a different split.  We assume that we have pixel-wise object-annotations to use to produce these counterfactuals by either removing or adding objects.  As the reviewer noted, this might not be sufficient for all spurious features (eg, high frequency signals).
>
> Next, we have emphasized, throughout  the paper, that SPIRE assumes that we have “access to pixel-wise object-annotations” for our specific experiments.
>
> Finally, we have added Appendix B.1, which goes through the types of problems SPIRE can be applied to and how to do so.  Importantly, it defines the image splits in more general terms.  As the reviewer noted, the definition of the image splits depend on both the model’s task and some other spurious feature.
>
> We hope that better explaining these core concepts in Section 3 will help make Section 3.1 easier to understand.
>
>
> **“No Object Annotation Experiment”**
> We have clarified this experiment by adding pseudo-code for and a discussion of the projection algorithm to Appendix H.1.  Since the goal of this projection is to add or remove the most “obvious” signal for the presence of spurious from the representation, we define “success” as having changed the linear model's prediction;  the algorithm that we use is guaranteed to do this.  Note that “removing the most obvious signal” is not the same as “removing all information,”  but the latter may be an interesting direction for future work.

---

### Review · Reviewer_tKuP · 2022-06-29

**Summary Of Contributions:**

This paper proposes a framework for identifying and repairing spurious correlations given object-level bounding box annotations (towards the end of the paper it also explores identifying the correlations without explicit annotations). The main idea is---given a "main" object that is causally related to the label and a "spurious" object that is non-causally related to the label---to view the dataset as a distribution over images containing the main object, the spurious object, both objects, or neither. This allows the authors to formally define a spurious correlation, and use image-level counterfactuals to find them. Finally, the paper repairs these correlations by counterfactually augmenting the data, while also being careful to remove any potential correlation between the augmentation artifacts and the label.

**Requested Changes:**

- Show example counterfactual images generated by SPIRE (beyond just Figure 1)
- Clarification on the composition of the subgroups
- Reporting subgroup accuracy instead of gaps [Critical]
- Error bars for Figure 4 and other results [Critical]
- Precise definition of what the "balanced dataset" used for evaluation is [Critical]
- Better structuring and/or worked example to improve understanding
- More precise comparison to prior works---many of the works cited as comparable are not actually about identifying and mitigating spurious annotated objects (for example, Xiao et al studies only the effect of backgrounds on classification). Perhaps a table that describes what each baseline method studies, what assumptions it makes, and what outcomes it measures would be useful.

**Strengths And Weaknesses:**

Overall, I thought this was a solid contribution to the study of spurious correlations in object detection. The reliance on explicit object annotations for the majority of the paper is a limitation, as is the inability of the method to distinguish what is truly a "spurious" correlation, but these both seem like reasonable limitations for the problem being studied, with the latter being more of a philosophical limitation than a technical one.

### Strengths:

- The categorization of images into Main, Spurious, Both, and Neither subgroups is very simple but still powerful in that it lets the paper formally define many of the concepts it seeks to detect and mitigate.
- The secondary objective of the mitigation step (de-correlating the augmentation itself from the label) is clever and missing from many of the spurious correlation mitigation papers I have seen.
- More generally, I appreciate the shift in perspective from identifying spurious correlations on a given image to identifying them across the entire dataset.

### Weaknesses:

- Requiring object-level annotations is a rather major limitation---while it does not limit the significance of the work in my opinion, it should be more prominently featured (reading the introduction one would guess that the paper automatically discovers the tennis racket-human association, when in practice one can "detect" this correlation by just counting co-occurence between people and tennis rackets using the bounding box annotations). In particular, the "identification" part of SPIRE seems to be more like "verification" of biases that are rather easy to detect by just counting co-occurence frequencies.
- Presenting the performance in terms of gaps can be misleading, since one way to make the gap zero is to degrade performance on both subgroups. At least in the Appendix there should be a plot showing the accuracy on each subgroup as P(Main|Spurious) is varied. In my opinion, this plot should replace Figure 4 (b) and (c) in the paper as these make it difficult to tell what the tradeoffs SPIRE is making are.
- The graphs are all missing error bars---given the (seemingly) highly random nature of SPIRE (since it is primarily implemented using data augmentation), it seems worth having error bars around, e.g., Figure 4(a).
- The paper was a bit confusing to follow---the writing was clear but the organization could be significantly improved. My suggestion would be to work through the tennis racket example fully in the introduction or shortly afterwards: show example images from each subgroup, show the counterfactual images and results from the identification step of SPIRE, then show what data augmentation would look like in this setting, then show subgroup accuracies before and after applying both SPIRE and one baseline method.

### Questions and Suggestions:

- Does the "only spurious" subgroup contain images from classes other than the class of interest? E.g., in the tennis racket example, would a person cooking be an example of an image in the "only spurious" subgroup?
- On page 6, the paper says "As a result, the gap metrics and the performance on the balanced distribution also only use natural images." This makes sense for the gap metrics, but my understanding was that the balanced distribution is constructed via counterfactual augmentation. If so, how does the evaluation account for the fact that SPIRE has seen counterfactually augmented images while the baseline methods have not? If not, might there be biases introduced by, e.g., differences between the distribution of the different subgroups?
- Are there cases where counting co-occurence with a label does *not* find a spurious correlation, but SPIRE does? (e.g., where the presence of spurious object A is not correlated with label X in terms of count) but a counterfactual analysis reveals a dependence? Similarly, are there instances where there *is* a co-occurence bias but a counterfactual analysis suggests that the model does not actually use this correlation? If so, this seems worth mentioning in the main text of the paper.
- The paper briefly mentions that the baseline methods make the same assumptions, but this is a bit vague. Does this just mean they also assume access to annotated bounding boxes?

---

> ### Author Response · Authors · 2022-07-03
> **Main Response (Part 1)**
>
> **“Show example counterfactual images generated by SPIRE (beyond just Figure 1)”**
> Additional examples are in Figure 2 and Figure 5 in Appendix B.2.  Because SPIRE uses the same types of counterfactuals as past work, we have made the reference to Appendix B.2 clearer.
>
> **“Clarification on the composition of the subgroups”**
> In the main paper, we have clarified that any object other than Main or Spurious can be present or absent without consequence for any of the subgroups.  Further, we think that Table 8 in Appendix B.1, which we added, should help clarify this as well.
>
>
> **“Reporting subgroup accuracy instead of gaps [Critical]”**
> The reviewer’s assessment that it is possible to shrink the gap metrics to zero by only decreasing the model’s accuracy on each subgroup is correct. However, we note that this strategy would result in a lower performance on the balanced distribution (e.g. Balanced Accuracy in Figure 4). We expanded the existing discussion on the “relationship between performance on the balanced distribution and the gap metrics” in Appendix C.3 to include this strategy.
>
> Additionally, please note that both Figure 6, in Appendix D, and Figure 10, in Appendix F, show the subgroup accuracies.
>
> **“Error bars for Figure 4 and other results [Critical]”**
> We have added a discussion on “Measuring Variance Across Trials” to Section 5 and added the standard deviation of the change in Average Precision between the mitigated and the baseline model to Tables 3-7.
>
>  Each value in Figure 4 is averaged over 64 measurements (8 object pairs with 8 trials each), and the standard error of these values is very low. We didn’t add error bars because they made the figure too crowded.  However, by comparing Figures 4 and 11, we can see that SPIRE has a consistent effect between two independently run sets of experiments
>
> **“Precise definition of what the ‘balanced dataset’ used for evaluation is [Critical]”**
> We have revised Section 4 accordingly.  During training, SPIRE produces the balanced distribution by augmenting the training distribution with counterfactual images.  During evaluation, we estimate a model’s performance on the balanced distribution by re-weighting its accuracy on each of image splits (where the per-split accuracies are calculated using exclusively natural images from the test set).  Note that, in the Section 2, we discuss methods that mitigate spurious patterns using a similar re-weighting strategy.
>
> **“Better structuring and/or worked example to improve understanding”**
> We thank the reviewer for this suggestion. We tried to provide an overview in Figure 1, without requiring the reader to grasp the details of how we split the subgroups. We want to improve the clarity of the paper (and hopefully have already done so in this revision), but are unsure of how to incorporate this suggestion without overwhelming the reader with an array of concepts that are presented in detail in Section 3, and are needed to understand the augmentation strategy.
>
> **“More precise comparison to prior works”**
> Could the reviewer elaborate on which works are inadequately described?  In general, we aimed to discuss works that relate to spurious patterns in image models more broadly than the specific version of the problem studied in this paper:
> -  Not all of them directly address the “spurious pattern” problem.  For example, Domain Adaptation and Group Distributionally Robust Optimization are not directly intended to address “spurious patterns” but they potentially could be used to do so.
> -  Those that do address the “spurious pattern” problem may not focus on “spurious objects”.  For example, Xiao et al focus on spurious patterns that are in the background of an image.
> -  They don’t all discuss both “identifying” and “mitigating”.
> -  Some focus on image classification or object detection or caption generation.
>
> Would including an explanation along these lines address this concern?

---

> ### Author Response · Authors · 2022-07-03
> **Main Response (Part 2)**
>
> **Clarify SPIRE’s Assumptions (“Requiring object-level annotations is a rather major limitation---while it does not limit the significance of the work in my opinion, it should be more prominently featured” & “The paper briefly mentions that the baseline methods make the same assumptions, but this is a bit vague”)**
>
> This was a common suggestion and one that we have acted upon in three ways:
> -  Throughout the paper, we have emphasized that SPIRE assumes that we have “access to pixel-wise object-annotations” for our specific experiments.
> -  At the end of the “Preliminaries” paragraph in Section 3, we have summarized the two general assumptions that must be satisfied in order to use SPIRE.
> -  We have added Appendix B.1, which goes through the process defining what types of problems SPIRE can be applied to and how to do so.
>
>
> **Co-occurrence vs Counterfactuals (“the ‘identification’ part of SPIRE seems to be more like ‘verification’ of biases that are rather easy to detect by just counting co-occurence frequencies.” & “Are there cases where counting co-occurence with a label does not find a spurious correlation, but SPIRE does?” & “Similarly, are there instances where there is a co-occurence bias but a counterfactual analysis suggests that the model does not actually use this correlation?”)**
>
>
> SPIRE does find spurious patterns that aren’t the result of co-occurrence, but not very many.  The “bias” column of Table 2 shows that SPIRE finds (Toothbrush, Person) and (Bird, Sheep) where the (Main, Spurious) objects are not correlated.
>
> SPIRE does reject Spurious Patterns that should exist based on Co-occurrence.  In the Benchmark Experiments (Section 5.1), we note that a strong positive correlation (ie, co-occurence) is not sufficient for a model to learn the implied Spurious Pattern for 5 out of the 13 object pairs; SPIRE’s counterfactual analysis usually agrees with this conclusion (this is in Figure 8 in Appendix F, which we added).  In the Full Experiment (Section 5.2), we note that many of the ~2700 object pairs that SPIRE rejected are strongly correlated (eg, baseball-bat and baseball-glove are strongly correlated but show almost no counterfactual relationship).
>
> We discuss this in the second paragraph of the “Identification” heading of Appendix A, but have also added a shortened version to the Conclusion.

---

### Author Response · Authors · 2022-07-03
**Major changes to the paper are marked in blue text**

We thank the reviewers for their comments and have updated the draft of the paper accordingly.

---

### Decision · Action_Editors · 2022-08-09

**Recommendation:** Accept with minor revision

**Comment:**

This paper proposes an approach for identifying spurious patterns in a training set when pixel-wise object-annotations are available, and a data-augmentation method for removing such patterns. The reviewers generally liked the paper, but had concerns about the clarity of the presentation. Reviewers also thought that the requirement of having pixel-wise object-annotations available was a major weakness.

The authors have responded to these concerns and improved the presentation as well as making other improvements requested by the reviewers. The reviewers seemed to be satisfied by these changes and recommended acceptance.

While the authors have clarified in the main text that their approach requires the availability of pixel-wise object-annotations, I believe that it is important to also mention this in the abstract. I recommend acceptance conditioned on the minor revision of making this clear in the abstract.